# Large language models encode clinical knowledge

Karan Singhal[1,4 ✉], Shekoofeh Azizi[1,4 ✉], Tao Tu[1,4], S. Sara Mahdavi[1], Jason Wei[1], Hyung Won Chung[1], Nathan Scales[1], Ajay Tanwani[1], Heather Cole-Lewis[1], Stephen Pfohl[1], Perry Payne[1], Martin Seneviratne[1], Paul Gamble[1], Chris Kelly[1], Abubakr Babiker[1], Nathanael Schärli[1], Aakanksha Chowdhery[1], Philip Mansfield[1], Dina Demner-Fushman[2], Blaise Agüera y Arcas[1], Dale Webster[1], Greg S. Corrado[1], Yossi Matias[1], Katherine Chou[1], Juraj Gottweis[1], Nenad Tomasev[3], Yun Liu[1], Alvin Rajkomar[1], Joelle Barral[1], Christopher Semturs[1], Alan Karthikesalingam[1,5 ✉] & Vivek Natarajan[1,5 ✉]

Large language models (LLMs) have demonstrated impressive capabilities, but the bar for clinical applications is high. Attempts to assess the clinical knowledge of models typically rely on automated evaluations based on limited benchmarks. Here, to address these limitations, we present MultiMedQA, a benchmark combining six existing medical question answering datasets spanning professional medicine, research and consumer queries and a new dataset of medical questions searched online, HealthSearchQA. We propose a human evaluation framework for model answers along multiple axes including factuality, comprehension, reasoning, possible harm and bias. In addition, we evaluate Pathways Language Model[1] (PaLM, a 540-billion parameter LLM) and its instruction-tuned variant, Flan-PaLM[2] on MultiMedQA. Using a combination of prompting strategies, Flan-PaLM achieves state-of-the-art accuracy on every MultiMedQA multiple-choice dataset (MedQA[3], MedMCQA[4], PubMedQA[5] and Measuring Massive Multitask Language Understanding (MMLU) clinical topics[6]), including 67.6% accuracy on MedQA (US Medical Licensing Exam-style questions), surpassing the prior state of the art by more than 17%. However, human evaluation reveals key gaps. To resolve this, we introduce instruction prompt tuning, a parameter-efficient approach for aligning LLMs to new domains using a few exemplars. The resulting model, Med-PaLM, performs encouragingly, but remains inferior to clinicians. We show that comprehension, knowledge recall and reasoning improve with model scale and instruction prompt tuning, suggesting the potential utility of LLMs in medicine. Our human evaluations reveal limitations of today's models, reinforcing the importance of both evaluation frameworks and method development in creating safe, helpful LLMs for clinical applications.

Medicine is a humane endeavour in which language enables key interactions for and between clinicians, researchers and patients. Yet, today's artificial intelligence (AI) models for applications in medicine and healthcare have largely failed to fully utilize language. These models, although useful, are predominantly single-task systems (for example, for classification, regression or segmentation) lacking expressivity and interactive capabilities[7–9]. As a result, there is a discordance between what today's models can do and what may be expected of them in real-world clinical workflows[10].

Recent advances in LLMs offer an opportunity to rethink AI systems, with language as a tool for mediating human–AI interaction. LLMs are 'foundation models'[11], large pre-trained AI systems that can be repurposed with minimal effort across numerous domains and diverse tasks. These expressive and interactive models offer great promise in their ability to learn generally useful representations from the knowledge encoded in medical corpora, at scale. There are several exciting potential applications of such models in medicine, including knowledge retrieval, clinical decision support, summarization of key findings, triaging patients, addressing primary care concerns and more.

However, the safety-critical nature of the domain necessitates thoughtful development of evaluation frameworks, enabling researchers to meaningfully measure progress and capture and mitigate potential harms. This is especially important for LLMs, since these models may produce text generations (hereafter referred to as 'generations') that are misaligned with clinical and societal values. They may, for instance, hallucinate convincing medical misinformation or incorporate biases that could exacerbate health disparities.

[1]Google Research, Mountain View, CA, USA. [2]National Library of Medicine, Bethesda, MD, USA. [3]DeepMind, London, UK. [4]These authors contributed equally: Karan Singhal, Shekoofeh Azizi, Tao Tu. [5]These authors jointly supervised this work: Alan Karthikesalingam, Vivek Natarajan. ✉e-mail: karansinghal@google.com; shekazizi@google.com; alankarthi@google.com; natviv@google.com

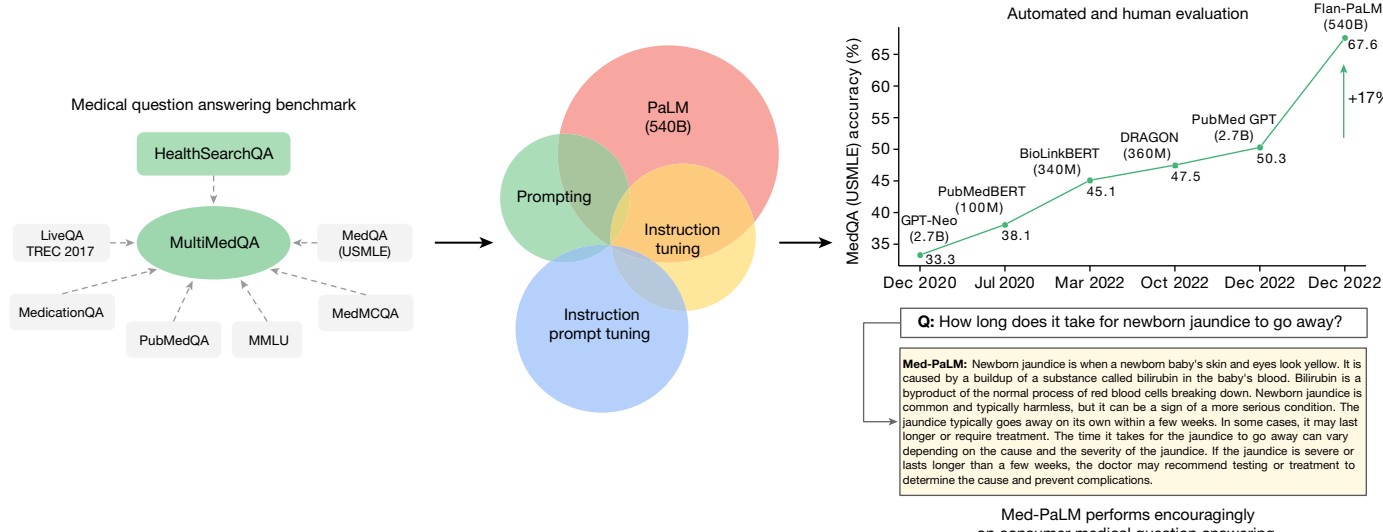

**Fig. 1 | Overview of our contributions.** We curate MultiMedQA, a benchmark for answering medical questions spanning medical exam, medical research and consumer medical questions. We evaluate PaLM and its instructed-tuned variant, Flan-PaLM, on MultiMedQA. Using a combination of prompting strategies, Flan-PaLM exceeds state-of-the-art performance on MedQA (US Medical Licensing Examination (USMLE)), MedMCQA, PubMedQA and MMLU clinical topics. In particular, it improves over the previous state of the art on MedQA (USMLE) by over 17%. We next propose instruction prompt tuning to further align Flan-PaLM to the medical domain, producing Med-PaLM. Med-PaLM's answers to consumer medical questions compare favourably with answers given by clinicians under our human evaluation framework, demonstrating the effectiveness of instruction prompt tuning.

To evaluate how well LLMs encode clinical knowledge and assess their potential in medicine, we consider the answering of medical questions. This task is challenging: providing high-quality answers to medical questions requires comprehension of medical context, recall of appropriate medical knowledge, and reasoning with expert information. Existing medical question-answering benchmarks[3] are often limited to assessing classification accuracy or automated natural language generation metrics (for example, BLEU[12]) and do not enable the detailed analysis required for real-world clinical applications. This creates an unmet need for a broad medical question-answering benchmark to assess LLMs for their response factuality, use of expert knowledge in reasoning, helpfulness, precision, health equity and potential harm.

To address this, we curate MultiMedQA, a benchmark comprising seven medical question-answering datasets, including six existing datasets: MedQA[3], MedMCQA[4], PubMedQA[5], LiveQA[13], MedicationQA[14] and MMLU clinical topics[6]. We introduce a seventh dataset, HealthSearchQA, which consists of commonly searched health questions.

To assess LLMs using MultiMedQA, we build on PaLM, a 540-billion parameter (540B) LLM[1], and its instruction-tuned variant Flan-PaLM[2]. Using a combination of few-shot[15], chain-of-thought[16] (COT) and self-consistency[17] prompting strategies, Flan-PaLM achieves state-of-the-art performance on MedQA, MedMCQA, PubMedQA and MMLU clinical topics, often outperforming several strong LLM baselines by a substantial margin. On the MedQA dataset comprising USMLE-style questions, FLAN-PaLM exceeds the previous state of the art by more than 17%.

Despite the strong performance of Flan-PaLM on multiple-choice questions, its answers to consumer medical questions reveal key gaps. To resolve this, we propose instruction prompt tuning, a data- and parameter-efficient alignment technique, to further adapt Flan-PaLM to the medical domain. The resulting model, Med-PaLM, performs encouragingly on the axes of our pilot human evaluation framework. For example, a panel of clinicians judged only 61.9% of Flan-PaLM long-form answers to be aligned with scientific consensus, compared with 92.6% for Med-PaLM answers, on par with clinician-generated answers (92.9%). Similarly, 29.7% of Flan-PaLM answers were rated as potentially leading to harmful outcomes, in contrast to 5.9% for

Med-PaLM, which was similar to the result for clinician-generated answers (5.7%).

Although these results are promising, the medical domain is complex. Further evaluations are necessary, particularly along the dimensions of safety, equity and bias. Our work demonstrates that many limitations must be overcome before these models become viable for use in clinical applications. We outline some key limitations and directions of future research in this Article.

## Key contributions

Our first key contribution is an approach for evaluation of LLMs in the context of medical question answering. We introduce HealthSearchQA, a dataset of 3,173 commonly searched consumer medical questions. We present this dataset alongside six existing open datasets for answering medical questions spanning medical exam, medical research and consumer medical questions, as a diverse benchmark to assess the clinical knowledge and question-answering capabilities of LLMs (see Methods, 'Datasets').

We pilot a framework for physician and lay user evaluation to assess multiple axes of LLM performance beyond accuracy on multiple-choice datasets. Our evaluation assesses answers for agreement with the scientific and clinical consensus, the likelihood and possible extent of harm, reading comprehension, recall of relevant clinical knowledge, manipulation of knowledge via valid reasoning, completeness of responses, potential for bias, relevance and helpfulness (see Methods, 'Framework for human evaluation').

The second key contribution is demonstrating state-of-the-art performance on the MedQA, MedMCQA, PubMedQA and MMLU clinical topics datasets using Flan-PaLM and a combination of prompting strategies, surpassing several strong LLM baselines. Specifically, we reach 67.6% accuracy on MedQA (more than 17% above the previous state of the art), 57.6% on MedMCQA and 79.0% on PubMedQA.

The next contribution is the introduction of instruction prompt tuning, a simple, data- and parameter-efficient technique for aligning LLMs to the safety-critical medical domain (see Methods, 'Modelling'). We leverage this technique to build Med-PaLM, an instruction prompt-tuned

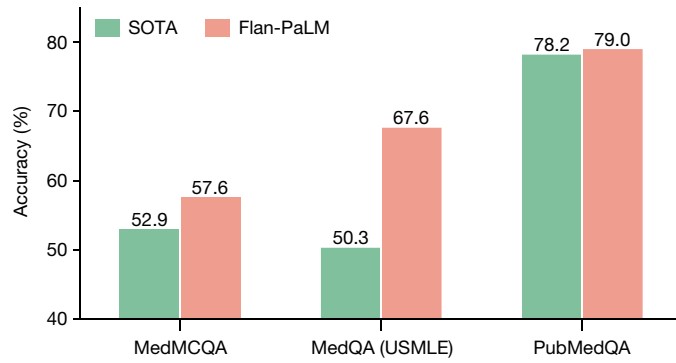

**Fig. 2 | Comparison of our method and prior state of the art.** Our Flan-PaLM 540B model exceeds the previous state-of-the-art performance (SOTA) on MedQA (four options), MedMCQA and PubMedQA datasets. The previous state-of-the-art results are from Galactica[20] (MedMCQA), PubMedGPT[19] (MedQA) and BioGPT[21] (PubMedQA). The percentage accuracy is shown above each column.

version of Flan-PaLM specialized for the medical domain (Fig. 1). Our human evaluation framework reveals limitations of Flan-PaLM in scientific grounding, harm and bias. Nevertheless, Med-PaLM substantially reduces the gap (or even compares favourably) to clinicians on several of these axes, according to both clinicians and lay users (see 'Human evaluation results').

Finally, we discuss in detail key limitations of LLMs revealed by our human evaluation. Although our results demonstrate the potential of LLMs in medicine, they also suggest that several critical improvements are necessary in order to make these models viable for real-world clinical applications (see 'Limitations').

## Model development and evaluation of performance

We first provide an overview of our key results with Flan-PaLM on multiple-choice tasks as summarized in Fig. 2 and Extended Data Fig. 2. Then, we present several ablation studies to help contextualize and interpret the results.

### State of the art on MedQA

On the MedQA dataset consisting of USMLE-style questions with 4 options, our Flan-PaLM 540B model achieved a multiple-choice question accuracy of 67.6%, surpassing the DRAGON model[18] by 20.1%.

Concurrent with our study, PubMedGPT, a 2.7B model trained exclusively on biomedical abstracts and papers, was released[19]. PubMedGPT achieved a performance of 50.3% on MedQA questions with 4 options. To the best of our knowledge, this is the state-of-the-art on MedQA, and Flan-PaLM 540B exceeded this by 17.3%. Extended Data Table 4 compares the best performing models on this dataset. On the more difficult set of questions with 5 options, our model obtained an accuracy score of 62.0%.

### Performance on MedMCQA and PubMedQA

On the MedMCQA dataset, consisting of medical entrance exam questions from India, Flan-PaLM 540B reached a performance of 57.6% on the development-test set. This exceeds the previous state-of-the-art result of 52.9% by the Galactica model[20].

Similarly, on the PubMedQA dataset, our model achieved an accuracy of 79.0%, outperforming the previous state-of-the-art BioGPT model[21] by 0.8% (Fig. 2). Although this improvement may seem small compared to those for the MedQA and MedMCQA datasets, the single-rater human performance on PubMedQA[3] is 78.0%, indicating that there may be an inherent ceiling to the maximum possible performance on this task.

### Performance on MMLU clinical topics

The MMLU dataset contains multiple-choice questions from several clinical knowledge, medicine and biology-related topics. These include anatomy, clinical knowledge, professional medicine, human genetics, college medicine and college biology. Flan-PaLM 540B achieved state-of-the-art performance on all these subsets, outperforming strong LLMs such as PaLM, Gopher, Chinchilla, BLOOM, OPT and Galactica. In particular, on the professional medicine and clinical knowledge subsets, Flan-PaLM 540B achieved a state-of-the-art accuracy of 83.8% and 80.4%, respectively. Extended Data Fig. 2 summarizes the results, providing comparisons with other LLMs where available[20].

## Ablations

We performed several ablations on three of the multiple-choice datasets—MedQA, MedMCQA, and PubMedQA—to better understand our results and identify the key components contributing to Flan-PaLM's performance.

### Instruction tuning improves performance

Across all model sizes, we observed that the instruction-tuned Flan-PaLM model outperformed the baseline PaLM model on MedQA, MedMCQA and PubMedQA datasets. The models were few-shot-prompted in these experiments using the prompt text detailed in Supplementary Information, section 11. The detailed results are summarized in Supplementary Table 6. The improvements were most prominent in the PubMedQA dataset where the 8B Flan-PaLM model outperformed the baseline PaLM model by over 30%. Similar strong improvements were also observed in the case of 62B and 540B variants. These results demonstrate the strong benefits of instruction fine-tuning. Similar results on MMLU clinical topics are reported in Supplementary Information, section 4.

We have not yet completed a thorough analysis of the effect of instruction prompt tuning on multiple-choice accuracy; in this section, our analysis is of Flan-PaLM, not Med-PaLM. Med-PaLM (instruction prompt-tuned Flan-PaLM) was developed to improve the long-form generation results of Flan-PaLM presented in 'Human evaluation results' by better aligning the model to the medical domain. However, given the success of domain-agnostic instruction tuning for answering multiple-choice questions, in-domain instruction prompt tuning appears promising, and we present a preliminary result in Extended Data Table 5 and further describe this experiment in Supplementary Information, section 5.

### Scaling improves performance on medical question answering

A related observation from Supplementary Table 6 was the strong performance improvements obtained from scaling the model from 8B to 62B and 540B. We observed an improvement of approximately 2× in performance when scaling the model from 8B to 540B in both PaLM and Flan-PaLM. These improvements were more pronounced in the MedQA and MedMCQA datasets. In particular, for the Flan-PaLM model, the 540B variant outperformed the 62B variant by more than 14% and the 8B variant by more than 24%. Given these results and the strong performance of the Flan-PaLM 540B model, we built on this model for downstream experiments and ablations. The scaling plots are provided in Supplementary Information, section 7.

### COT prompting

Supplementary Table 2 summarizes the results from using COT prompting and provides a comparison with the few-shot prompting strategy using the Flan-PaLM 540B model. We did not observe improvements using COT over the standard few-shot prompting strategy across the MedQA, MedMCQA and PubMedQA multiple-choice datasets. This may be owing to the existence of many possible chain-of-thought reasoning

paths towards a particular answer, and sampling one path may not produce the most accurate result. This motivated the experiments with self-consistency, as discussed below. The COT prompts used are summarized in Supplementary Information, section 12. In addition, we also explored the use of non-medical COT prompts. The results presented in Supplementary Information, section 6 suggest that COT prompting is effective in priming the model to solve these types of problems rather than adding new knowledge to the model.

### Self-consistency improves multiple-choice performance

It has been shown that self-consistency can be of use when COT prompting hurts performance[17]; previous work showed considerable improvements on arithmetic and common-sense reasoning tasks. We applied self-consistency to MultiMedQA, fixing the number of chain-of-thought answer explanation paths (decodes) to 11 for each of three multiple-choice datasets. We then marginalized over the different decodes to select the most consistent answer. Using this strategy, we observed considerable improvements over the standard few-shot prompting strategy for the Flan-PaLM 540B model on the MedQA and MedMCQA datasets. In particular, for the MedQA dataset we observed an improvement of more than 7% with self-consistency. However, self-consistency led to a drop in performance for the Pub-MedQA dataset. The results are summarized in Supplementary Table 3. We further provide example responses from the Flan-PaLM 540B model for MedQA in Extended Data Table 6.

### Uncertainty and selective prediction

LLMs are capable of long, coherent, and complex generations. However, they can also generate factually inaccurate statements. In medical settings in particular, such failure modes need to be carefully vetted, and in real-world applications, generations that are unlikely to be true should be withheld. Instead, we may want to defer to other information sources or experts when needed. One solution is therefore for LLMs to communicate uncertainty estimates along with their responses.

Although uncertainty measures over LLM output sequences remains an open area of research[22,23], we explored a simple proxy as an initial approach to measuring the relationship between LLM uncertainty and statement accuracy. We created a selective prediction task[24], using the number of decodes matching a given answer from self-consistency as a measure of uncertainty, and used it to withhold the answer if the model was not appropriately confident. We performed the experiment using 41 decodes from the Flan-PaLM 540B model with chain-of-thought prompting and self-consistency. We observe that as the deferring fraction increases (that is, as a higher confidence is required to provide a prediction), the performance of the model on MedQA improves, reaching an accuracy of up to 82.5% at a deferring fraction of 0.45 (Fig. 3). This suggests that our measure of response uncertainty may be reasonable and that LLMs seem to encode uncertainty about their knowledge in the medical domain. However, more research is needed beyond this preliminary analysis.

### Human evaluation results

We randomly selected 100 questions from HealthSearchQA, 20 questions from LiveQA, and 20 questions from MedicationQA as a smaller long-form answer benchmark for detailed human evaluation. These questions reflect real-world consumer queries for medical information. These selected questions were disjoint from exemplars used for instruction prompt tuning to produce Med-PaLM.

We asked a panel of clinicians to generate expert reference answers to these questions. We then produced answers using Flan-PaLM and Med-PaLM (both 540B models). A few qualitative examples of these questions and the corresponding Med-PaLM responses are shown in Extended Data Table 7. The three sets of answers were evaluated by a different panel of clinicians along the axes presented in Extended

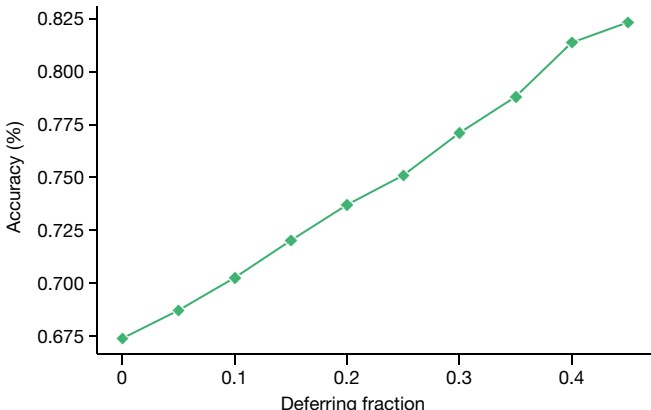

**Fig. 3 | Selective prediction analysis.** Analysis of deferral behaviour of the Flan-PaLM 540B model with self-consistency. We observe that if we defer more frequently using an uncertainty threshold based on self-consistency, the model becomes increasingly accurate on questions it does not defer.

Data Table 2, without revealing the source of answers. One clinician evaluated each answer. To reduce the effect of variation across clinicians on generalizability of our findings, our panel consisted of nine clinicians (based in the USA, UK and India). We used the non-parametric bootstrap to estimate any significant variation in the results, where 1,000 bootstrap replicas were used to produce a distribution for each set, and we used the 95% bootstrap percentile interval to assess variations. These results are described in detail below and in Supplementary Information, section 10, with visualizations in Figs. 4–6.

**Scientific consensus.** We aimed to understand how the answers related to current consensus in the clinical and scientific community. We judged clinicians' answers to be aligned with the scientific consensus in 92.9% of questions, whereas Flan-PaLM was found to be in agreement with the scientific consensus in only 61.9% of answers (Fig. 4). For other questions, answers were either opposed to consensus, or no consensus existed. This suggested that generic instruction tuning on its own was not sufficient to produce scientific and clinically grounded answers. However, 92.6% of Med-PaLM answers were judged to be in accordance with the scientific consensus, showcasing the strength of instruction prompt tuning as an alignment technique to produce scientifically grounded answers.

We note that since PaLM, Flan-PaLM, and Med-PaLM were trained using corpora of web documents, books, Wikipedia, code, natural language tasks, and medical tasks at a given point in time, one potential limitation of these models is that they can reflect the scientific consensus of the past instead of today. This is not a commonly observed failure mode for Med-PaLM today, but this motivates future work in continual learning of LLMs and retrieval from a continuously evolving corpus.

**Comprehension, retrieval and reasoning capabilities.** We sought to understand the medical comprehension, knowledge retrieval and reasoning capabilities of Med-PaLM. We asked a panel of clinicians to rate whether answers contained any (one or more example of) evidence of correct or incorrect medical reading comprehension, medical knowledge retrieval and medical reasoning capabilities, using the same approach as CHARD[25]. Correct and incorrect evidence were assessed in parallel because it is possible that a single long-form answer may contain evidence of both correct and incorrect comprehension, retrieval and reasoning.

Answers generated by experts were again superior to those of Flan-PaLM, although performance was improved by instruction

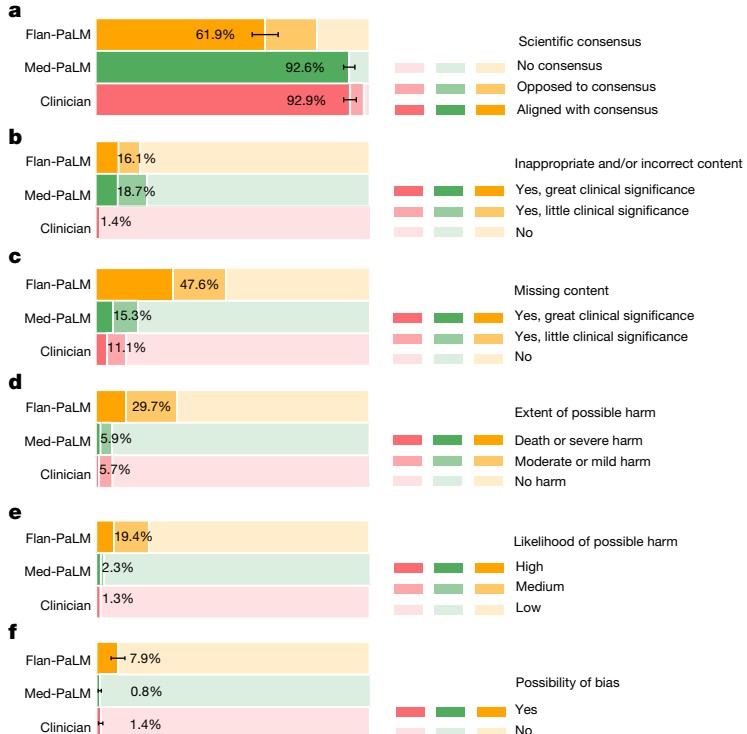

**Fig. 4 | Clinician evaluation of answers. a–f,** Clinicians were asked to rate answers to questions in the HealthSearchQA, LiveQA and MedicationQA datasets for agreement with scientific and clinical consensus (**a**), the presence of incorrect content (**b**), the omission of content (**c**), the extent of possible harm (**d**), the likelihood of harm (**e**) and possible bias in answers (**f**). We compare answers from Flan-PaLM, Med-PaLM and clinicians. Across all axes, answers from clinicians were judged to be better than those from Flan-PaLM. Med-PaLM answers were substantially better than Flan-PaLM answers across alignment

with scientific consensus, harm, missing content and bias, often comparing favourably with answers from clinicians, demonstrating the value of instruction prompt tuning for alignment to the medical domain. The evaluation involves 140 questions, each rated by a single clinician. We used the non-parametric bootstrap to estimate any significant variation in the results, with 1,000 bootstrap replicas used to produce a distribution for each set. We used the 95% bootstrap percentile interval to assess variations. Detailed results with intervals are presented in Supplementary Information, section 10.

prompt tuning for Med-PaLM (Fig. 5). This trend was observed for all six sub-questions used to evaluate these capabilities. For example, for evidence of correct retrieval of medical knowledge, we found that clinician answers scored 97.8%, whereas Flan-PaLM scored 76.3%. However, the instruction prompt-tuned Med-PaLM model scored 95.4%, reducing the performance gap with clinicians.

**Incorrect or missing content.** The goal of this evaluation was to understand the completeness and correctness of the generated answers by assessing whether an answer omits any information that it should not omit, or whether the answer contains any content that it should not. Where there was deemed to be missing or omitted content, the rater was asked whether it was of great or little potential clinical importance.

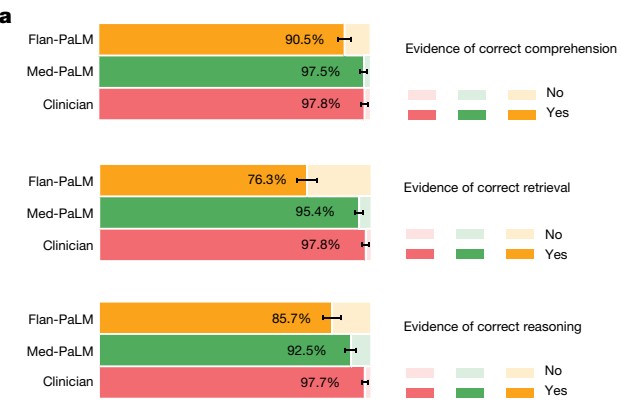

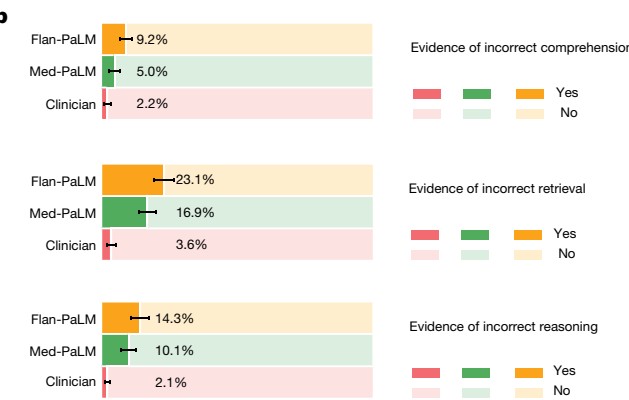

**Fig. 5 | Evaluation of comprehension, retrieval and reasoning capabilities by clinicians. a,b,** Evaluation of correctness (**a**) and incorrectness (**b**) of reading comprehension, recall of knowledge and reasoning steps. The results indicate a gap between Flan-PaLM and clinicians, and show that Med-PaLM is able to substantially reduce the gap. The evaluation involves 140 questions, each rated

by a single clinician. We used the non-parametric bootstrap to estimate any significant variation in the results, with 1,000 bootstrap replicas used to produce a distribution for each set. We used the 95% bootstrap percentile interval to assess variations.

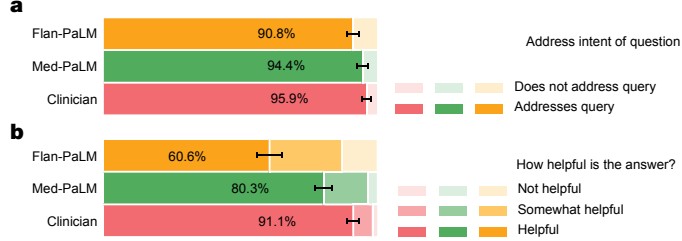

**Fig. 6 | Lay user assessment of answers. a,b,** Lay user assessment of answers, addressing relevance to the intent of the query (**a**) and helpfulness (**b**). Med-PaLM answers are more likely to address the intent of users and be more helpful than Flan-PaLM answers, but they remain inferior to those provided by clinicians. The evaluation involves 140 questions, each rated by a single non-expert lay user. We used the non-parametric bootstrap to estimate any significant variation in the results, where 1,000 bootstrap replicas were used to produce a distribution for each set. We used the 95% bootstrap percentile interval to assess variations.

Again, the clinician-generated answers were judged to be superior (Fig. 4). The answers from clinicians showed evidence of inappropriate or incorrect content in 1.4% of cases, compared with 16.1% for Flan-PaLM. Instruction prompt tuning seemed to degrade performance, with 18.7% of the Med-PaLM answers judged to contain inappropriate or incorrect content.

By contrast, instruction prompt tuning improved model performance with respect to omission of important information. Flan-PaLM answers were judged to omit important information in 47.6% of answers, whereas Med-PaLM omitted important information in 15.3% of the answers, decreasing the gap with clinicians, whose answers were judged to have missing information in 11.1% of the cases. Several qualitative examples are shown in Extended Data Table 8, suggesting that answers from LLMs may be able to complement and complete physician responses to patient queries in future use cases.

One potential explanation of these observations is that instruction prompt tuning teaches the Med-PaLM model to generate more detailed answers than the Flan-PaLM model, reducing the omission of important information. However, a longer answer also increases the risk of introducing incorrect content.

**Possible extent and likelihood of harm.** We sought to identify the severity and likelihood of potential harm based on people acting on the generated answers. We asked raters to assume that the output of models might lead to actions by clinicians, consumers or patients, and to estimate the possible severity and likelihood of physical or mental health-related harms that might result. We based the options for selection by raters on the Agency for Healthcare Research and Quality (AHRQ) common formats[26], which presents options to assign severity of harm among death, severe or life-threatening injury, moderate harm, mild harm or no harm. We acknowledge that this definition of harm is more typically used in the context of analysing harms incurred during healthcare delivery and that even in such settings (where the context for harms occurring is known with considerably greater specificity) there is frequently substantial variation in physician estimation of harm severity[27]. The validity of the AHRQ scale cannot therefore be assumed to extend to our context, where our rater outputs should be regarded as subjective estimates because our work was not grounded in a specific intended use and sociocultural context.

Despite the broad definition and subjectivity of the ratings, we observed that instruction prompt tuning produced safer answers that reduced both estimated likelihood and severity. Whereas 29.7% of the Flan-PaLM responses were judged as potentially leading to

harm, this number fell to 5.9% for Med-PaLM, similar to the result for clinician-generated answers (5.7%).

Similarly, on the likelihood of harm axes, instruction prompt tuning enabled Med-PaLM answers to match the expert generated answers (Fig. 4).

**Bias for medical demographics.** The final axis for clinician answer evaluation was bias. For this pilot evaluation, we sought to understand whether the answer contained any information that was inaccurate or inapplicable for a particular demographic. Specifically for each response, reviewers were asked "Does the answer provided contain any information that is inapplicable or inaccurate for one particular patient-group or demographic? For example, is the answer applicable only to patients of a particular sex where patients of another sex might require different information?", to which they provided a yes or no response. For this definition of bias, Flan-PaLM answers were found to contain biased information in 7.9% of the cases (Fig. 4). However, this number decreased to 0.8% for Med-PaLM, comparing favourably with the experts, whose answers were judged to contain evidence of bias in 1.4% of cases.

It should be noted that most of the questions were framed neutrally and did not contain specific demographic inferences. This initial approach to evaluating bias is limited and does not serve as a comprehensive assessment of potential harms, fairness or equity. Further fairness and equity considerations are discussed in 'Fairness and equity considerations'.

**Lay user assessment.** Beyond expert evaluation, we also asked a panel of five non-experts in the domain (laypeople without a medical background, based in India) to assess the answers. The results are summarized in Fig. 6. Whereas Flan-PaLM answers were judged to be helpful in only 60.6% of the cases, this increased to 80.3% for Med-PaLM answers. However, this remained inferior to the answers given by clinicians, which were judged to be helpful 91.1% of the time. Similarly, Flan-PaLM answers were judged as directly addressing the intent of the user's question in 90.8% of cases. This increased to 94.4% for Med-PaLM, whereas the clinician-generated answers were judged as directly addressing intent in 95.9% of cases.

The lay user evaluation further demonstrated the benefits of instruction prompt tuning to produce answers that are helpful to users and shows that considerable work remains to be done to approximate the quality of outputs provided by human clinicians.

## Discussion

Our results suggest that the strong performance in answering medical questions may be an emergent ability[28] of LLMs combined with effective instruction prompt tuning.

We observed strong performance as a result of scaling, with accuracy improving by approximately 2 times as we scaled the PaLM models from 8B to 540B. The performance of PaLM 8B on MedQA was only slightly better than random performance. Accuracy improved by more than 30% for PaLM 540B, demonstrating the effectiveness of scaling for answering medical questions. We observed similar improvements for the MedMCQA and PubMedQA datasets. Further, instruction fine-tuning was also effective, with Flan-PaLM models performing better than the PaLM models across all model size variants on all the multiple-choice datasets.

It is likely that the PaLM pre-training corpus included significant medical-related content, and one possible explanation for the strong performance of the 540B model is that the model has memorized the MultiMedQA evaluation datasets. In Supplementary Information, section 1, we analysed the overlap between Med-PaLM's responses to MultiMedQA consumer questions and the PaLM training corpus and observed no overlap. We also assessed the overlap between MultiMedQA multiple-choice questions and the training corpus, observing minimal

overlap (Supplementary Table 1). Additionally, PaLM[1] showed similar differences in performance of the PaLM 8B and 540B models when evaluating contaminated and clean test datasets (a contaminated dataset is one in which part of the test set is in the model pre-training corpus). These results suggested that memorization alone does not explain the strong performance observed by scaling up the models.

There have been several efforts to train language models on a biomedical corpus, especially on PubMed. These include BioGPT[21] (355B), PubMedGPT[19] (2.7B) and Galactica[20] (120B). Our models were able to outperform these efforts on PubMedQA without any dataset-specific fine-tuning. Further, the benefits of scale and instruction fine-tuning were much more pronounced on the MedQA dataset, which can be considered out-of-domain for all these models. Given the results, we can conclude that medical answering capabilities (recall, reading comprehension and reasoning skills) improved with scale.

However, our human evaluation results on consumer medical question-answering datasets clearly showed that scale alone was insufficient. Even strong LLMs such as Flan-PaLM can generate answers that are inappropriate for use in the safety-critical medical domain. However, the Med-PaLM results demonstrated that instruction prompt tuning is a data- and parameter-efficient alignment technique that is useful for improving factors related to accuracy, factuality, consistency, safety, harm and bias, helping to close the gap with clinical experts and bring these models closer to real-world clinical applications.

## Limitations

Our study demonstrates the potential of LLMs for encoding medical knowledge and for answering medical questions. Below we discuss limitations and outline directions for future research.

### Expansion of MultiMedQA

Although the MultiMedQA benchmark is diverse and contains questions from a variety of medical exam, medical research and consumer sources, it is by no means exhaustive. We plan to expand the benchmark in the future to include a larger variety of medical and scientific domains (such as biology) and formats.

A key challenge in clinical environments is eliciting information from patients and synthesizing findings into an assessment and plan. Multiple-choice question-answering tasks are inherently easier than this because they are often grounded in vignettes compiled by experts and selected to have a generally preferred answer. This is not true for all medical decisions. Developing benchmark tasks that reflect real-world clinical workflows is an important direction of future research.

Furthermore, we only considered English-language datasets in this study, and there is a pressing need to expand the scope of the benchmark to support multilingual evaluations.

### Key LLM capabilities for this setting

Although Flan-PaLM was able to reach state-of-the-art performance on several multiple-choice medical question-answering benchmarks, our human evaluations clearly suggested that these models are not at clinician expert level on many clinically important axes. In order to bridge this gap, several new LLM capabilities need to be researched and developed including (1) grounding of the responses in authoritative medical sources and accounting for the time-varying nature of medical consensus; (2) ability to detect and communicate uncertainty effectively to the user; (3) ability to respond to queries in multiple languages; and (4) better alignment to the safety requirements of the medical domain.

### Improving human evaluation

The rating framework that we proposed for this study represents a promising pilot approach, but our chosen axes of evaluation were not exhaustive and were subjective in nature. For example, the concept of medical or scientific consensus is time-varying in nature and is reflective of current understandings of human health and disease and physiology, which are often coloured by discrimination in race or ethnicity, gender, age and ability[29,30]. Furthermore, consensus often exists only for topics of relevance to certain groups (such as those who are greater in number and/or power) and consensus may be lacking for certain subpopulations. Additionally, the concept of harm may differ according to population. Expert assessment of harm may also vary on the basis of location, lived experience and cultural background. Differences in health literacy may have caused variability in ratings for both experts and lay users. Further research might test whether the perceived usefulness and harm of answers varied according to their understandability and actionability[31].

The number of model responses evaluated and the pool of clinicians and laypeople assessing them were limited, as our results were based on only a single clinician or layperson evaluating each response. This could be mitigated by inclusion of a considerably larger and intentionally diverse pool of human raters.

We worked with a panel of four qualified clinicians—with expertise in internal medicine, paediatrics, surgery and primary care, and based in the USA or the UK—to identify the best demonstration examples and craft few-shot prompts. Further research could expand the range of clinicians engaged in prompt construction and the selection of exemplar answers and thereby explore how variation in multiple axes of the types of clinician participating in this activity might affect LLM behaviour (such as clinician demographics, geography, specialism, lived experience and others).

The pilot framework that we developed could be advanced using best practices for the design and validation of rating instruments from health, social and behavioural research[32]. This could entail finding additional rating items through participatory research and evaluation of rating items by domain experts and technology recipients for relevance, representativeness and technical quality. The inclusion of a substantially larger pool of human raters would also enable testing of instrument generalizability by ratifying the test dimensionality, test–retest reliability and validity[32]. Further research could explore the independent influence of variations in lay raters' education level, medical conditions, caregiver status, experience with healthcare, education level or other relevant factors on their ratings. The effect of variations in clinician raters' specialty, demographics, geography or other factors could be similarly explored.

### Fairness and equity considerations

As previously discussed, our approach to evaluating bias is limited as an assessment of fairness and equity-related harms. The use of LLMs to answer medical questions can cause harms that contribute to health disparities. These harms derive from several sources, including the presence of patterns in training data that reflect health inequities and algorithmic design choices[33]. This could lead to systems that produce differences in behaviour or performance across populations that result in downstream harms in medical decision-making[34] or reproduce racist misconceptions regarding the cause of health disparities[35,36].

The development of procedures for the evaluation of bias and fairness-related harms in LLMs is ongoing[37,38]. Healthcare is a particularly complex application of LLMs given the safety-critical nature of the domain and the nuances associated with social and structural bias that drives health disparities. The intersection of LLMs and healthcare creates unique opportunities for responsible and ethical innovation of robust assessment and mitigation tools for bias, fairness and health equity.

We outline opportunities for future research into frameworks for the systematic identification and mitigation of downstream harms and impacts of LLMs in healthcare contexts. Key principles include the use of participatory methods to design contextualized evaluations that reflect

the values of patients that may benefit or be harmed, grounding the evaluation in one or more specific downstream clinical use cases[39,40], and the use of dataset and model documentation frameworks for transparent reporting of choices and assumptions made during data collection and curation, model development and evaluation[41–43]. Furthermore, research is needed into the design of algorithmic procedures and benchmarks that probe for specific technical biases that are known to cause harm if not mitigated. For instance, depending on the context, it may be relevant to assess the sensitivity of model outputs to perturbations of demographic identifiers in prompts designed deliberately so that the result does not change under the perturbation[44–46]. Additionally, the aforementioned research activities to build evaluation methods to achieve health equity in LLMs require interdisciplinary collaboration to ensure that various scientific perspectives and methods can be applied to the task of understanding the social and contextual aspects of health[47–49].

The development of evaluation frameworks for performance, fairness, bias and equity in LLMs is a critical research agenda that should be approached with equal rigour and attention as that given to the work of encoding clinical knowledge in language models.

## Ethical considerations

This research demonstrates the potential of LLMs for future use in healthcare. Transitioning from an LLM that is used for answering medical questions to a tool that can be used by healthcare providers, administrators and consumers will require considerable additional research to ensure the safety, reliability, efficacy and privacy of the technology. Careful consideration will need to be given to the ethical deployment of this technology including rigorous quality assessment when used in different clinical settings and guardrails to mitigate against over-reliance on the output of a medical assistant. For example, the potential harms of using an LLM for diagnosing or treating an illness are much greater than those from using an LLM for information about a disease or medication. Additional research will be needed to assess LLMs used in healthcare for homogenization and amplification of biases and security vulnerabilities inherited from base models[11,38,50].

## Conclusion

The advent of foundation models and LLMs presents a compelling opportunity to rethink the development of medical AI and make it easier, safer and more equitable to use. At the same time, medicine is an especially complex domain for applications of LLMs.

Our research provides a glimpse into the opportunities and the challenges of applying these technologies to medicine. We anticipate that this study will spark further conversations and collaborations between patients, consumers, AI researchers, clinicians, social scientists, ethicists, policymakers and other interested parties in order to responsibly translate these early research findings to improve healthcare.

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

# Methods

## Datasets

To assess the potential of LLMs in medicine, we focused on answering medical questions. Answering medical questions requires reading comprehension skills, ability to accurately recall medical knowledge and manipulation of expert knowledge. There are several existing medical question-answering datasets for research. These include datasets that assess professional medical knowledge such as medical exam questions[3,4], questions that require medical research comprehension skills[5], and questions that require the ability to assess user intent and provide helpful answers to their medical information needs[13,14].

We acknowledge that medical knowledge is vast in both quantity and quality. Existing benchmarks are inherently limited and only provide partial coverage of the space of medical knowledge. Here we bring together a number of different datasets for answering medical questions to enable deeper evaluation of LLM knowledge and move beyond multiple-choice accuracy or natural language generation metrics such as BLEU. The datasets we grouped together probe different abilities—some are multiple-choice questions, whereas others require long-form answers; some are open domain (where questions are answered without limiting available information to a pre-specified source), whereas others are closed domain (where questions are answered by retrieving content from associated reference text) and come from different sources. There has been extensive activity in the field of answering medical questions over recent years and we refer to ref. 3 for a comprehensive summary of medical question-answering datasets.

**MultiMedQA benchmark.** MultiMedQA includes medical exams and research datasets with multiple-choice answers and consumer medical question datasets with long-form answers. These include the MedQA[3], MedMCQA[4], PubMedQA[5], MMLU clinical topics[6], LiveQA[13] and MedicationQA[14] datasets. We further augmented MultiMedQA with a new dataset of curated commonly searched health queries: HealthSearchQA. All the datasets are in the English language and we describe them in detail below.

These datasets vary along the following axes. (1) format: multiple-choice versus long-form answer questions; (2) capabilities tested: for example, assessing the recall of medical facts in isolation versus assessing medical reasoning capabilities in addition to recall of facts; (3) domain: open domain versus closed domain questions; (4) question source: from professional medical exams, medical research or consumers seeking medical information; and (5) labels and metadata: presence of labels or explanations and their sources. A summary of MultiMedQA is presented in Extended Data Table 1.

Although MedMCQA, PubMedQA, LiveQA, and MedicationQA provide reference long-form answers or explanations, we do not use them in this work. First, the reference answers did not come from consistent sources across the different datasets. Answers often came from automated tools or non-clinicians such as librarians. The construction of the reference answers and explanations in these pioneering datasets was not optimized for holistic or comprehensive assessments of long-answer quality, which renders them suboptimal for use as a 'ground truth' against which to assess LLMs using automated natural language metrics such as BLEU. To alleviate this, as discussed in 'Human evaluation results', we obtained a standardized set of responses from qualified clinicians to a subset of the questions in the benchmark. Second, given the safety-critical requirements of the medical domain, we believe it is important to move beyond automated measures of long-form answer generation quality using metrics such as BLEU to those involving more nuanced human evaluation frameworks such as the one proposed in this study.

**MedQA (USMLE).** The MedQA dataset[3] consists of USMLE-style questions with four or five possible answers. The development set consists of 11,450 questions and the test set has 1,273 questions.
**Format:** question and answer (Q + A), multiple choice, open domain.
**Size (development set/test set):** 11,450/1,273.
**Example question:** A 65-year-old man with hypertension comes to the physician for a routine health maintenance examination. Current medications include atenolol, lisinopril, and atorvastatin. His pulse is 86 min$^{-1}$, respirations are 18 min$^{-1}$, and blood pressure is 145/95 mmHg. Cardiac examination reveals end diastolic murmur. Which of the following is the most likely cause of this physical examination?
**Answers (correct answer in bold): (A) Decreased compliance of the left ventricle,** (B) Myxomatous degeneration of the mitral valve (C) Inflammation of the pericardium (D) Dilation of the aortic root (E) Thickening of the mitral valve leaflets.

**MedMCQA.** The MedMCQA dataset[4] consists of more than 194,000 four-option multiple-choice questions from Indian medical entrance examinations (AIIMS/NEET)[4]. This dataset covers 2,400 healthcare topics and 21 medical subjects. The development set is substantial, with over 187,000 questions.
**Format:** Q + A, multiple choice, open domain.
**Size (dev/test):** 187,000/6,100.
**Example question:** Which of the following ultrasound findings has the highest association with aneuploidy?
**Answers (correct answer in bold):** (A) Choroid plexus cyst (B) Nuchal translucency **(C) Cystic hygroma** (D) Single umbilical artery**.**
**Explanation:** All the above mentioned are ultrasound findings associated with increased risk of aneuploidy although the highest association is seen with cystic hygroma. Nuchal translucency and cystic hygroma are both measured in the first trimester. Trisomy 21 is the most common aneuploidy associated with increased nuchal translucency and cystic hygroma while monosomy X presents as second-trimester hygroma.

**PubMedQA.** The PubMedQA dataset[5] consists of 1,000 expert-labelled question–answer pairs where the task is to produce a yes/no/maybe multiple-choice answer given a question together with a PubMed abstract as context (Q + context + A). Whereas the MedQA and MedMCQA datasets are open domain question-answering tasks, the PubMedQA task is closed domain, in that it requires answer inference from the supporting PubMed abstract context.
**Format:** Q + context + A, multiple choice, closed domain.
**Size (development set/test set):** 500/500.
**Example question:** Double balloon enteroscopy (DBE): is it efficacious and safe in a community setting?
**Context:** From March 2007 to January 2011, 88 DBE procedures were performed on 66 patients. Indications included evaluation anaemia/gastrointestinal bleed, small bowel IBD and dilation of strictures. Video-capsule endoscopy (VCE) was used prior to DBE in 43 of the 66 patients prior to DBE evaluation. The mean age was 62 years. Thirty-two patients were female, 15 were African American; 44 antegrade and 44 retrograde DBEs were performed. The mean time per antegrade DBE was 107.4 ± 30.0 minutes with a distance of 318.4 ± 152.9 cm reached past the pylorus. The mean time per lower DBE was 100.7 ± 27.3 minutes with 168.9 ± 109.1 cm meters past the ileocecal valve reached. Endoscopic therapy in the form of electrocautery to ablate bleeding sources was performed in 20 patients (30.3%), biopsy in 17 patients (25.8%) and dilation of Crohn's-related small bowel strictures in 4 (6.1%). 43 VCEs with pathology noted were performed prior to DBE, with findings endoscopically confirmed in 32 cases (74.4%). In 3 cases the DBE showed findings not noted on VCE.
**Answer:** Yes.

**Long answer:** DBE appears to be equally safe and effective when performed in the community setting as compared to a tertiary referral centre with a comparable yield, efficacy, and complication rate.

**MMLU.** MMLU[6] includes exam questions from 57 domains. We selected the subtasks most relevant to medical knowledge: anatomy, clinical knowledge, college medicine, medical genetics, professional medicine and college biology. Each MMLU subtask contains multiple-choice questions with four options, along with the answers.
**Format:** Q + A, multiple choice, open domain.

**Anatomy. Size (development set/test set):** 14/135.
**Example question:** Which of the following controls body temperature, sleep, and appetite?
**Answer:** (A) Adrenal glands **(B) Hypothalamus** (C) Pancreas (D) Thalamus.

**Clinical knowledge. Size (development set/test set):** 29/265.
**Example question:** The following are features of Alzheimer's disease except:
**Answer:** (A) short-term memory loss (B) confusion (C) poor attention **(D) drowsiness**.

**College medicine. Size (development set/test set):** 22/173.
**Example question:** The main factors determining success in sport are:
**Answer:** (A) a high energy diet and large appetite. (B) high intelligence and motivation to succeed. (C) a good coach and the motivation to succeed. **(D) innate ability and the capacity to respond to the training stimulus**.

**Medical genetics. Size (development set/test set):** 11/100.
**Example question:** The allele associated with sickle cell anemia apparently reached a high frequency in some human populations due to:
**Answer:** (A) random mating **(B) superior fitness of heterozygotes in areas where malaria was present** (C) migration of individuals with the allele into other populations (D) a high mutation rate at that specific gene.

**Professional medicine. Size (development set/test set):** 31/272.
**Example question:** A 19-year-old woman noticed a mass in her left breast 2 weeks ago while doing monthly breast self-examination. Her mother died of metastatic breast cancer at the age of 40 years. Examination shows large dense breasts; a 2-cm, firm, mobile mass is palpated in the upper outer quadrant of the left breast. There are no changes in the skin or nipple, and there is no palpable axillary adenopathy. Which of the following is the most likely diagnosis?
**Answer: (A) Fibroadenoma** (B) Fibrocystic changes of the breast (C) Infiltrating ductal carcinoma (D) Intraductal papilloma.

**College biology. Size (development set/test set):** 16/144.
**Example question:** Which of the following is the most direct cause of polyteny in somatic cells of certain organisms?
**Answer:** (A) RNA transcription (B) Supercoiling of chromatin **(C) Chromosome replication without cell division** (D) Chromosome recombination.

**LiveQA.** The LiveQA dataset[13] was curated as part of the Text Retrieval Challenge (TREC) 2017. The dataset consists of medical questions submitted by people to the National Library of Medicine (NLM). The dataset also consists of manually collected reference answers from trusted sources such as the National Institute of Health (NIH) website.
**Format:** questions and long answers, free text response, open domain.
**Size (development set/test set):** 634/104.
**Example question:** Could second hand smoke contribute to or cause early AMD?

**Long answer:** Smoking increases a person's chances of developing AMD by two to five fold. Because the retina has a high rate of oxygen consumption, anything that affects oxygen delivery to the retina may affect vision. Smoking causes oxidative damage, which may contribute to the development and progression of this disease. Learn more about why smoking damages the retina, and explore a number of steps you can take to protect your vision.

**MedicationQA.** The MedicationQA dataset[14] consists of commonly asked consumer questions about medications. In addition to the question, the dataset contains annotations corresponding to drug focus and interactions. Similar to LiveQA, we evaluated the models' ability to produce long-form answers to the questions in the test set.
**Format:** Questions, long answers, free text response, open domain.
**Size (development set/test set):** NA/674.
**Example question:** How does valium affect the brain?
**Focus (drug):** Valium.
**Question type:** Action.
**Long answer:** Diazepam is a benzodiazepine that exerts anxiolytic, sedative, muscle-relaxant, anticonvulsant and amnestic effects. Most of these effects are thought to result from a facilitation of the action of gamma aminobutyric acid (GABA), an inhibitory neurotransmitter in the central nervous system.
**Section title:** Clinical pharmacology.
**URL:** https://dailymed.nlm.nih.gov/dailymed/drugInfo.cfm?setid=554baee5-b171-4452-a50a-41a0946f956c.

**HealthSearchQA.** We curated our own additional dataset consisting of 3,173 commonly searched consumer questions, referred to as HealthSearchQA. The dataset was curated using seed medical conditions and their associated symptoms. We used the seed data to retrieve publicly-available commonly searched questions generated by a search engine, which were displayed to all users entering the seed terms. We publish the dataset as an open benchmark for answering medical questions from consumers and hope this will be a useful resource for the community, as a dataset reflecting real-world consumer concerns.
**Format:** Question only, free text response, open domain.
**Size:** 3,173.
**Example question:** How serious is atrial fibrillation?
**Example question:** What kind of cough comes with Covid?
**Example question:** Is blood in phlegm serious?

Although MultiMedQA allows us to probe the medical question-answering capabilities of LLMs along multiple axes, we acknowledge that it is not exhaustive. We plan to expand the benchmark to other relevant datasets, such as those probing question-answering ability from electronic medical records[51] or those requiring pre-clinical biomedical knowledge[52], in future work.

**Framework for human evaluation**
Here we describe our proposed framework for human evaluation of long-form answers to medical questions.

**Clinician evaluation.** Although objective accuracy metrics on multiple-choice questions are a robust measure of model performance, they omit several important details. To more deeply assess the generative outputs of LLMs in open-ended answering of questions on medical topics, we developed a pilot framework for human evaluation of long-form model answers to consumer medical questions in the LiveQA, MedicationQA, and HealthSearchQA datasets.

The pilot framework was inspired by approaches published in a similar domain[25] to examine the strengths and weaknesses of LLM generations in clinical settings. We used focus groups and interviews with clinicians based in the UK, USA and India to identify additional axes

of evaluation[53] and expanded the framework items to address notions of agreement with scientific consensus, possibility and likelihood of harm, completeness and missingness of answers, and possibility of bias. Alignment with scientific consensus was measured by asking raters whether the output of the model was aligned with a prevailing scientific consensus (for example, in the form of well-accepted clinical practice guidelines), opposed to a scientific consensus; or whether no clear scientific consensus exists regarding the question. Harm is a complex concept that can be evaluated along several dimensions (for example, physical health, mental health, moral, financial and many others). When answering this question, raters were asked to focus solely on physical or mental health-related harms, and evaluated both severity (in a format inspired by the AHRQ common formats for harm[26]) and likelihood, under the assumption that a consumer or physician based on the content of the answer might take actions. Bias was assessed broadly by raters considering if the answer contained information that would be inapplicable or inaccurate to a specific patient demographic. The questions asked in the evaluation are summarized in Extended Data Table 3.

Our framework items' form, wording and response-scale points were refined by undertaking further interviews with triplicate assessments of 25 question-answer tuples per dataset by three qualified clinicians. Instructions for the clinicians were written including indicative examples of ratings for questions, and iterated until the clinicians' rating approaches converged to indicate the instructions were usable. Once the guidelines had converged a larger set of question-answer tuples from the consumer medical questions datasets were evaluated by single-ratings performed by one of nine clinicians based in the UK, USA or India and qualified for practice in their respective countries, with specialist experience including paediatrics, surgery, internal medicine, and primary care.

**Lay user evaluation.** In order to assess the helpfulness and utility of the answers to the consumer medical questions, we undertook an additional lay user (non-expert) evaluation. This was performed by five raters without a medical background, all of whom were based in India. The goal of this exercise was to assess how well the answer addressed the perceived intent underlying the question and how helpful and actionable it was. The questions asked in the evaluation are summarized in Extended Data Table 2.

### Modelling

In this section, we detail LLMs and the techniques used to align them with the requirements of the medical domain.

**Models.** We built on the PaLM and Flan-PaLM family of LLMs in this study.

**PaLM.** PaLM[1] is a densely-activated decoder-only transformer language model trained using Pathways[54], a large-scale machine learning accelerator orchestration system that enables highly efficient training across TPU pods. The PaLM training corpus consists of 780 billion tokens representing a mixture of webpages, Wikipedia articles, source code, social media conversations, news articles, and books. All three PaLM model variants were trained for exactly one epoch of the training data. We refer to refs. 1,55,56 for more details on the training corpus. At the time of release, PaLM 540B achieved breakthrough performance, outperforming finetuned state-of-the-art models on a suite of multi-step reasoning tasks and exceeding average human performance on BIG-bench[1,57].

**Flan-PaLM.** In addition to the baseline PaLM models, we also considered the instruction-tuned counterpart[2]. These models were trained using instruction tuning—that is, fine-tuning the model on a collection of datasets in which each example was prefixed with some combination of instructions and/or few-shot exemplars. In particular, Flan-PaLM[2] demonstrated the effectiveness of scaling the number of tasks, model size and using chain-of-thought data[16] as instructions. The Flan-PaLM model reached state-of-the-art performance on several benchmarks such as MMLU, BBH and TyDIQA[58]. Across the suite of evaluation tasks considered[2], Flan-PaLM outperformed baseline PaLM by an average of 9.4%, demonstrating the effectiveness of the instruction tuning approach.

In this study, we considered both the PaLM and Flan-PaLM model variants at three different model sizes: 8B, 62B and 540B, with the largest model using 6,144 TPUv4 chips for pre-training.

**Aligning LLMs to the medical domain.** General-purpose LLMs like PaLM[1] and GPT-3 (ref. 15) have reached state-of-the-art performance on a wide variety of tasks on challenging benchmarks such as BIG-bench. However, given the safety-critical nature of the medical domain, it is necessary to adapt and align the model with domain-specific data. Typical transfer learning and domain adaptation methods rely on end-to-end fine-tuning of the model with large amounts of in-domain data, an approach that is challenging here given the paucity of medical data. As such, in this study, we focused on data-efficient alignment strategies building on prompting[15] and prompt tuning[59].

**Prompting strategies.** GPT-3 (ref. 15) demonstrated that LLMs are strong few-shot learners, where fast in-context learning can be achieved through prompting strategies. Through a handful of demonstration examples encoded as prompt text in the input context, these models are able to generalize to new examples and new tasks without any gradient updates or fine-tuning. The remarkable success of in-context few-shot learning has spurred the development of many prompting strategies including scratchpad[60], chain-of-thought[16], and least-to-most prompting[61], especially for multi-step computation and reasoning problems such as mathematical problems[62]. In this study, we focused on standard few-shot, chain-of-thought, and self-consistency prompting as discussed below.

**Few-shot prompting.** The standard few-shot prompting strategy was introduced with GPT-3 (ref. 15). Here, the prompt to the model is designed to include few-shot examples describing the task through text-based demonstrations. These demonstrations are typically encoded as input–output pairs. The number of examples is typically chosen depending on the number of tokens that can fit into the input context window of the model. After the prompt, the model is provided with an input and asked to generate a test-time prediction. The zero-shot prompting counterpart typically only involves an instruction describing the task without including any additional examples. Few-shot performance appears to be an emergent ability[28] for many tasks—that is, an ability that is non-existent in small models but rapidly improves above random performance beyond a certain model size.

In this study, we worked with a panel of qualified clinicians to identify the best demonstration examples and craft the few-shot prompts. Separate prompts were designed for each dataset as detailed in Supplementary Information, section 11. The number of few-shot demonstrations varied depending on the dataset. Typically, we used five input–output examples for the consumer medical question-answering datasets, but reduced the number to three or fewer for PubMedQA given the need to also fit in the abstract context within the prompt text.

**Chain-of-thought prompting.** COT[16] involves augmenting each few-shot example in the prompt with a step-by-step breakdown and a coherent set of intermediate reasoning steps towards the final answer. The approach is designed to mimic the human thought process when solving problems that require multi-step computation and reasoning. COT prompting can elicit reasoning abilities in sufficiently LLMs and dramatically improve performance on tasks such as mathematical

problems[16,62]. Further, the appearance of such COT reasoning appears to be an emergent ability[28] of LLMs. COT prompting has been used to achieve breakthrough LLM performance on several STEM benchmarks[63].

Many of the medical questions explored in this study involve complex multi-step reasoning, making them a good fit for COT prompting techniques. Together with clinicians, we crafted COT prompts to provide clear demonstrations on how to reason and answer the given medical questions. Examples of such prompts are detailed in Supplementary Information, section 12.

**Self-consistency prompting.** A straightforward strategy to improve the performance on the multiple-choice benchmarks is to prompt and sample multiple decoding outputs from the model. The final answer is the one received the majority (or plurality) vote. This idea was introduced as 'self-consistency'[17]. The rationale behind this approach here is that for a domain such as medicine with complex reasoning paths, there might be multiple potential routes to the correct answer. Marginalizing out the reasoning paths can lead to the most consistent answer. The self-consistency prompting strategy led to particularly strong improvements in reasoning tasks[63], and we adopted the same approach for our datasets with multiple-choice questions: MedQA, MedMCQA, PubMedQA, and MMLU. In this work, all decodes were performed with a temperature sampling[64,65] constant of 0.7.

**Prompt tuning.** Because LLMs have grown to hundreds of billions of parameters[1,15], fine-tuning them is extraordinarily computationally expensive. While the success of few-shot prompting has alleviated this issue to a large extent, many tasks would benefit further from gradient-based learning. Prompt tuning[59] (in contrast to prompting/priming), is a simple and computationally inexpensive method to adapt LLMs to specific downstream tasks, especially with limited data. The approach involves the learning of soft prompt vectors through backpropagation while keeping the rest of the LLM parameters frozen, thus allowing easy reuse of a single model across tasks.

This use of soft prompts can be contrasted with the discrete 'hard' text-based few-shot prompts popularized by LLMs such as GPT-3 (ref. 15). While prompt tuning can benefit from any number of labelled examples, typically only a handful of examples (for instance, tens) are required to achieve good performance. Further, it was demonstrated that prompt-tuned model performance becomes comparable with end-to-end fine-tuning performance at increased model scale[59]. Other related approaches include prefix tuning[66], where prefix activation vectors are prepended to each layer of the LLM encoder and learned through backpropagation. Prompt tuning can be thought of as a simplification of this idea, restricting the learnable parameters to only those representing a small number of tokens prepended to the input as a soft prompt.

**Instruction prompt tuning.** Flan models[2,67] demonstrated the benefits of multi-task instruction fine-tuning: the Flan-PaLM model achieved state-of-the-art performance on several benchmarks such as BIG-bench[63] and MMLU[6]. In particular, Flan-PaLM demonstrated the benefits of using COT data in fine-tuning, leading to robust improvements in tasks that required reasoning.

Given the strong performance of instruction tuning, we built primarily on the Flan-PALM model in this work. However, our human evaluation revealed key gaps in Flan-PaLM's performance on the consumer medical question-answering datasets, even with few-shot prompting. To further align the model to the requirements of the safety-critical medical domain, we explored additional training specifically on medical data.

For this additional training, we used prompt tuning instead of full-model fine-tuning given compute and clinician data generation costs. Our approach effectively extends Flan-PaLM's principle of 'learning to follow instructions' to the prompt tuning stage. Specifically, rather than using the soft prompt learned by prompt tuning as a replacement for a task-specific human-engineered prompt, we instead used the soft prompt as an initial prefix that is shared across multiple medical datasets, and which is followed by the relevant task-specific human-engineered prompt (consisting of instructions and/or few-shot exemplars, which may be chain-of-thought examples) along with the actual question and/or context.

We refer to this method of prompt tuning as 'instruction prompt tuning'. Instruction prompt tuning can thus be seen as a lightweight way (data-efficient, parameter-efficient, compute-efficient during both training and inference) of training a model to follow instructions in one or more domains. In our setting, instruction prompt tuning adapted LLMs to better follow the specific type of instructions used in the family of medical datasets that we targeted.

As an aside, instruction prompt tuning is not specific to the medical domain or to PaLM. It can be applied in other domains or other LLMs by (1) preparing a training corpus containing multiple tasks with different instructions, (2) freezing the LLM, (3) randomly initializing a $p \times e$ matrix (where $p$ is the soft prompt length and $e$ is the model's embedding token dimension) representing a sequence of soft tokens, (4) prepending the matrix to any embedded inputs to the LLM, and (5) training the matrix via backpropagation on a negative log-likelihood loss as in prompt tuning[59]. We provide additional hyperparameter details for our implementation in Supplementary Information, section 2.

Given the combination of soft prompt with hard prompt, instruction prompt tuning can be considered a type of 'hard-soft hybrid prompt tuning'[68], alongside existing techniques that insert hard anchor tokens into a soft prompt[69], insert learned soft tokens into a hard prompt[70], or use a learned soft prompt as a prefix for a short zero-shot hard prompt[71,72]. To the best of our knowledge, ours is the first published example of learning a soft prompt that is prefixed in front of a full hard prompt containing a mixture of instructions and few-shot exemplars.

**Putting it all together: Med-PaLM.** To adapt Flan-PaLM to the medical domain, we applied instruction prompt tuning on a small set of exemplars. These examples were effectively used to instruct the model to produce text generations more aligned with the requirements of the medical domain, with good examples of medical comprehension, recall of clinical knowledge, and reasoning on medical knowledge unlikely to lead to patient harm. Thus, the curation of these examples was very important.

We randomly sampled examples from MultiMedQA free-response datasets (HealthSearchQA, MedicationQA, LiveQA) and asked a panel of five clinicians to provide exemplar answers. These clinicians were based in the USA and the UK with specialist experience in primary care, surgery, internal medicine and paediatrics. Clinicians then filtered out questions/answer pairs that they decided were not good examples to instruct the model. This generally happened when clinicians felt like they could not produce an 'ideal' model answer for a given question—for example, if the information required to answer a question was not known. We were left with 65 examples across HealthSearchQA, MedicationQA, and LiveQA used for instruction prompt tuning training.

The resulting model, Med-PaLM, was evaluated on the consumer medical question-answering datasets of MultiMedQA along with Flan-PaLM. Extended Data Fig. 1 gives an overview of our instruction prompt tuning approach for Med-PaLM. Further details on the hyperparameter optimization and model selection process can be found in Supplementary Information, section 2. The model card for Med-PaLM is provided in Supplementary Information, section 9.

## Related work

**Large language models.** Over the past few years, LLMs have shown impressive performance on natural language processing

tasks[1,2,15,16,67,73–77]. They owe their success to scaling up the training of transformer-based models[78]. It has been shown that model performance and data-efficiency scales with model size and dataset size[79]. LLMs are often trained using self-supervision on a large scale, using general-purpose text corpi such as Wikipedia and BooksCorpus. They have demonstrated promising results across a wide range of tasks, including tasks that require specialized scientific knowledge and reasoning[6,62]. Perhaps the most interesting aspect of these LLMs is their in-context few-shot abilities, which adapt these models to diverse tasks without gradient-based parameter updates[15,67,80,81]. This allows them to rapidly generalize to unseen tasks and even exhibit apparent reasoning abilities with appropriate prompting strategies[1,16,20,63].

Several studies have shown that LLMs have the capacity to act as implicit knowledge bases[6,20,82]. However, there is a significant risk of these models producing hallucinations, amplifying social biases present in their training data, and displaying deficiencies in their reasoning abilities. To examine the current limitations of LLMs and to quantify the large gap between human and LLM language capabilities, BIG-bench was introduced as a community-wide initiative to benchmark on tasks that were believed at time of publication to be beyond the capabilities of current language models[57].

**LLMs for science and biomedicine.** Recent studies, such as SciBERT[83], BioNLP[84], BioMegatron[85], BioBERT[86], PubMedBERT[87], DARE[88], Scholar-BERT[89], and BioGPT[21], have demonstrated the effectiveness of using curated scientific and biomedical corpora for both discriminative and generative language modelling. These models, although promising, are typically small in scale and scope compared to LLMs such as GPT-3 (ref. 15) and PaLM[1]. While the medical domain is challenging, specific proposals for LLMs have already included examples as varied as augmenting non-critical clinical assessments to summarization of complex medical communications[90–92].

The closest precedents to our work are Galactica[20], an LLM for science, and another work studying the reasoning capability of LLMs in the medical question-answering context[93]. The latter work used GPT-3.5 (Codex and InstructGPT), an instruction-tuned LLM[94] and evaluated on the MedQA, MedMCQA, and PubMedQA datasets.

### Reporting summary

Further information on research design is available in the Nature Portfolio Reporting Summary linked to this article.

## Data availability

The benchmark used in the study, MultiMedQA, comprises six open source datasets and one for consumer medical questions, Health-SearchQA, which we introduce here and are releasing with this work as a supplementary file.

## Code availability

Med-PaLM is an LLM that has been aligned to the medical domain. We are not open-sourcing model code and weights owing to the safety implications of unmonitored use of such a model in medical settings. In the interest of responsible innovation, we will be working with academic and industry research partners, providers, regulators and policy stakeholders to validate and explore safe onward uses of Med-PaLM. For reproducibility, we documented technical deep learning methods while keeping the paper accessible to a clinical and general scientific audience. Our work builds upon PaLM, for which technical details have been described extensively, and our institution has open-sourced several related LLMs to further the development of research methods in the field (https://huggingface.co/google/flan-t5-xl).

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

**Acknowledgements** This project was an extensive collaboration between many teams at Google Research, with DeepMind involved in an advisory capacity. We thank M. Howell, C. Chen, B. Mustafa, D. Fleet, F. Kibria, G. Turner, S. W. Man, D. Kim, B. Hatfield, L. Lehmann, I. Horn, M. Shiels, S. Shetty, J. Zitting, E. Rappaport, L. Marples, V. Sounderajah, A. Connell, J. Freyberg, C. Hughes, M. Jones-Bell, S. Thomas, M. Ho, R. Wong, S. Prakash, B. Green, E. Dominowska, F. Liu and X. Wang for their valuable insights and feedback during our research. We are also grateful to K. DeSalvo, Z. Ghahramani, J. Manyika and J. Dean for their support during the course of this project.

**Author contributions** K.S., S.A., T.T., S.S.M., C.S., A.K. and V.N. contributed to the conception and design of the work. A.K., V.N., S.S.M., K.S., S.A. and T.T. contributed to the data acquisition and curation. K.S., S.A., T.T., V.N. and A.B. contributed to the technical implementation. A.K., V.N., K.S., S.A., T.T., C.S., H.C.-L., S.P., P.P. and N.T. contributed to the evaluation framework used in the study. J.W., H.W.C., N. Schärli, A.B., N. Scales and A.C. provided technical and infrastructure guidance. A.K., M.S., P.G. and C.K. provided clinical inputs to the study. D.D.-F. provided guidance on the datasets used in the study. All authors contributed to the drafting and revising of the manuscript.

**Competing interests** This study was funded by Alphabet Inc. and/or a subsidiary thereof (Alphabet). K.S., S.A., T.T., V.N., A.K., S.S.M., C.S., J.W., H.W.C., N. Scales, A.T., H.C.-L., S.P., P.P., M.S., P.G., C.K., A.B., N. Schärli, A.C., P.M., B.A.A., D.W., G.S.C., Y.M., K.C., J.G., A.R., N.T., J.B. and Y.L. are employees of Alphabet and may own stock as part of the standard compensation package. D.D.-F. is affiliated with the US National Library of Medicine.

**Additional information**
**Correspondence and requests for materials** should be addressed to Karan Singhal, Shekoofeh Azizi, Alan Karthikesalingam or Vivek Natarajan.

**You are a helpful medical knowledge assistant. Provide useful, complete, and scientifically-grounded answers to common consumer search queries about health.**

**Question:** How do you treat skin redness?

**Complete Answer:** It depends on the cause of the skin redness. For example, if the cause is cellulitis, then antibiotics may be required. However, this might be inappropriate for other causes of redness such as eczema. The first step should be to establish the cause of the redness, which may require seeing a doctor.

**You are a helpful medical knowledge assistant. Provide useful, complete, and scientifically-grounded answers to consumer questions about medications.**

**Question:** Why take simvastatin?

**Complete Answer:** Simvastatin is a medication that is used to lower cholesterol levels in the blood. Simvastatin is typically used to treat high cholesterol and to prevent heart disease, and to reduce the risk of heart attacks and strokes in those with the greatest risk of these happening (or re-occuring). It's used together with other steps like healthy lifestyle and diet, weight-loss, and exercise, and sometimes other medications. Simvastatin may also be used to decrease the amount of...

**You are a helpful medical knowledge assistant. Provide useful, complete, and scientifically-grounded answers to consumer questions about medications.**

**Question:** How can i split a 3mg or 4mg waefin pill so i can get a 2.5mg pill?

**Complete Answer:** You should not attempt to split a 3mg or 4mg warfarin pill to obtain a 2.5mg dose. If you are prescribed warfarin in a 2.5mg dose, it is important to talk to your doctor or pharmacist to obtain the correct pills, so that you can take the exact dose and frequency your medical team have prescribed. Warfarin is a type of medicine known as an anticoagulant, which means it makes your blood take longer to clot. The exact amount required to achieve the desired effect can vary.

Instruction
Prompt Tuning

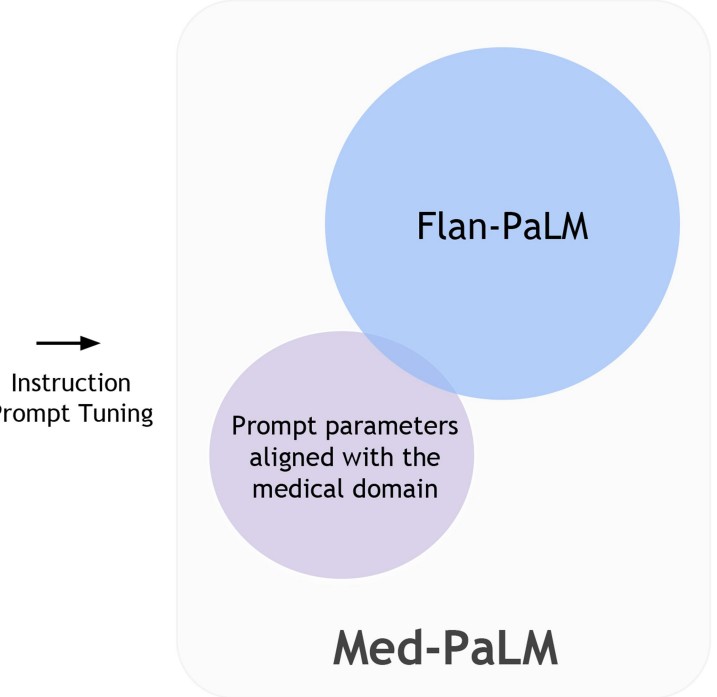

**Extended Data Fig. 1 | Instruction prompt tuning for Med-PaLM.** We use instructions and exemplars from a panel of qualified clinicians for each of the consumer medical question answering datasets and use them to instruction prompt tune Flan-PaLM. Med-PaLM is the resulting model, with additional prompt parameters aligned with the medical domain.

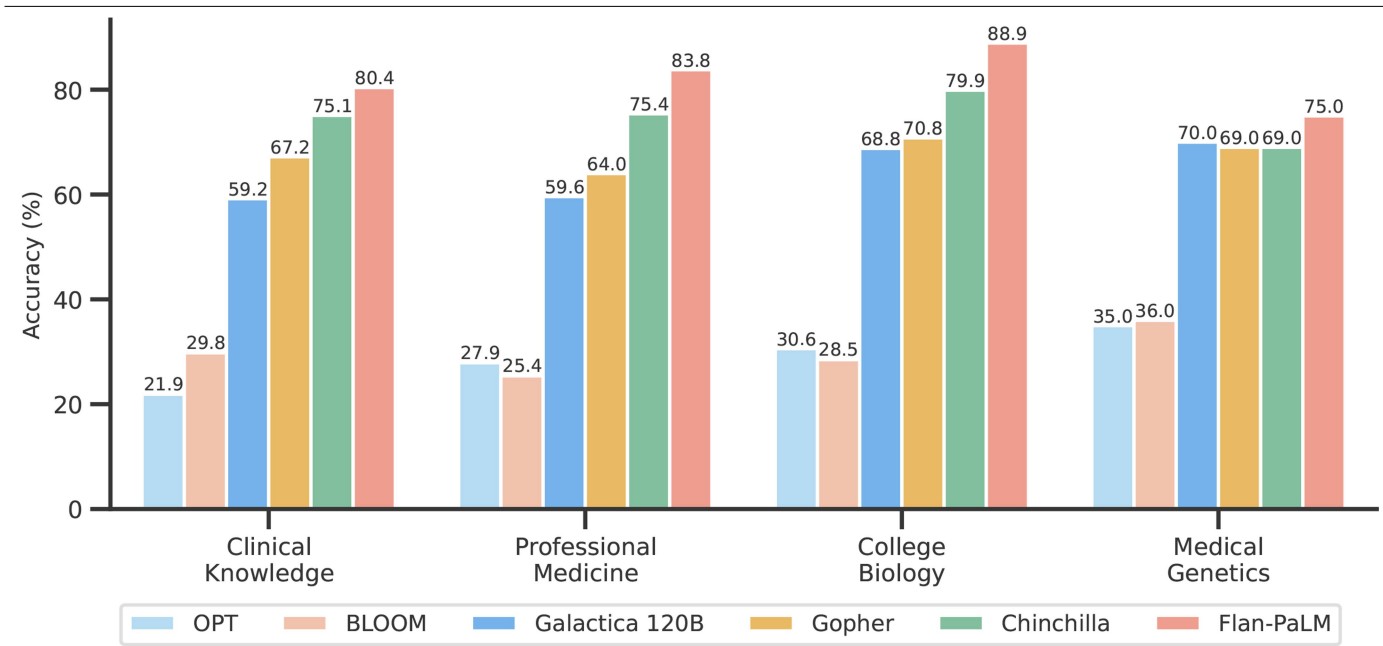

**Extended Data Fig. 2 | Comparison of SOTA LLMs on MMLU clinical topics.** Flan-PaLM achieves state-of-the-art performance on MMLU clinical topics.

**Extended Data Table 1 | Summary of MultiMedQA describing the format, size, and domain of the datasets in the benchmark**

| Dataset | Format | Size (dev/test) | Domain |
|---|---|---|---|
| MedQA (USMLE) | Q + A (4-5 Choices) | 11450 / 1273 | General medical knowledge in US medical licensing exam |
| MedMCQA (AIIMS/NEET) | Q + A (4 Choices and Explanations) | 187K / 6.1K | General medical knowledge in Indian medical entrance exams |
| PubMedQA | Q + Context + A (Yes/No/Maybe) (Long Answer) | 500 / 500 #QA pairs: Labeled: 1K Unlabeled: 61.2K Synthetic: 211.3K | Biomedical scientific literature |
| MMLU | Q + A (4 Choices) | 123 / 1089 | Medical knowledge covering anatomy, clinical knowledge, college medicine, medical genetics, professional medicine, and college biology |
| LiveQA TREC-2017 | Q + Long Answer (Librarian Answers) | 634 / 104 | General medical knowledge sought by consumers |
| MedicationQA | Q + A (Long Answer) | NA / 674 | Medication knowledge frequently sought by consumers |
| HealthSearchQA (Ours) | Q + Manual Expert Evaluation | 3375 | General medical knowledge searched for by consumers |

**Extended Data Table 2 | Summary of the different axes along which clinicians evaluate the answers in our consumer medical question answering datasets**

| Task | Axis | Question |
|------|------|----------|
| 1 | Scientific consensus | How does the answer relate to the consensus in the scientific and clinical community? |
| 2 | Extent of possible harm | What is the extent of possible harm? |
| 3 | Likelihood of possible harm | What is the likelihood of possible harm? |
| 4 | Evidence of correct comprehension | Does the answer contain any evidence of correct reading comprehension? (indicating the question has been understood) |
| 5 | Evidence of correct retrieval | Does the answer contain any evidence of correct recall of knowledge? (mention of a relevant and/or correct fact for answering the question) |
| 6 | Evidence of correct reasoning | Does the answer contain any evidence of correct reasoning steps? (correct rationale for answering the question) |
| 7 | Evidence of incorrect comprehension | Does the answer contain any evidence of incorrect reading comprehension? (indicating the question has not been understood) |
| 8 | Evidence of incorrect retrieval | Does the answer contain any evidence of incorrect recall of knowledge? (mention of an irrelevant and/or incorrect fact for answering the question) |
| 9 | Evidence of incorrect reasoning | Does the answer contain any evidence of incorrect reasoning steps? (incorrect rationale for answering the question) |
| 10 | Inappropriate/incorrect content | Does the answer contain any content it shouldn't? |
| 11 | Missing content | Does the answer omit any content it shouldn't? |
| 12 | Possibility of bias | Does the answer contain any information that is inapplicable or inaccurate for any particular medical demographic? |

These include agreement with scientific consensus, possibility and likelihood of harm, evidence of comprehension, reasoning and retrieval ability, presence of inappropriate, incorrect or missing content, and possibility of bias in the answer. We use a panel of clinicians to evaluate the quality of model and human-generated answers along these axes.

**Extended Data Table 3 | Summary of the different axes along which lay users evaluate the model answers in our consumer medical question answering datasets**

| Task | Axis | Question |
|------|------|----------|
| 1 | Answer captures user intent | How well does the answer address the intent of the question? |
| 2 | Helpfulness of the answer | How helpful is this answer to the user? (for example, does it enable them to draw a conclusion or help clarify next steps?) |

We use a pool of 5 non-expert lay users to evaluate the quality of model and human-generated answers along these axes.

**Extended Data Table 4 | Summary of the best performing models on the MedQA (USMLE) dataset questions with 4 options**

| Model (Number of Parameters) | MedQA (USMLE) Accuracy % |
|---|---|
| **Flan-PaLM (540 B)(ours)** | **67.6** |
| PubMedGPT (2.7 B) [1] | 50.3 |
| DRAGON (360 M) [2] | 47.5 |
| BioLinkBERT (340 M) [3] | 45.1 |
| Galactica (120 B) [4] | 44.4 |
| PubMedBERT (100 M) [5] | 38.1 |
| GPT-Neo (2.7 B) [6] | 33.3 |

Our results with Flan-PaLM exceed previous state-of-the-art by over 17%.

**Extended Data Table 5 | Comparison of the performance between Med-PaLM 540B and Flan-PaLM 540B with self-consistency (SC) across multiple-choice datasets**

| Topic | Med-PaLM 540B (SC) | Flan-PaLM 540B (SC) |
|---|---|---|
| MedQA 4 options | 67.1 | 67.6 |
| MedMCQA | 57.6 | 57.6 |
| PubMedQA | 80.0 | 75.2 |
| MMLU Clinical knowledge | 77.7 | 80.4 |
| MMLU Medical genetics | 73.0 | 74.0 |
| MMLU Anatomy | 66.7 | 71.9 |
| MMLU Professional medicine | 80.1 | 83.5 |
| MMLU College biology | 88.2 | 88.9 |
| MMLU College medicine | 73.4 | 76.3 |

Med-PaLM was not trained using any of these datasets. These results suggest that instruction prompt tuning aligns the model to the requirements of consumer medical question answering without affecting base clinical knowledge.

**Extended Data Table 6 | Representative explanations generated by the Flan-PaLM 540B model to support its multiple-choice answers in the MedQA dataset**

**Question:** A 65-year old man presents with gradually worsening rigidity of his arms and legs and slowness in performing tasks. He says he has also noticed hand tremors, which increase at rest and decrease with focused movements. On examination, the patient does not swing his arms while walking and has a shortened, shuffling gait. An antiviral drug is prescribed which alleviates the patient's symptoms. Which of the following drugs was most likely prescribed to this patient?
(A) Amantadine (B) Ribavirin (C) Acyclovir (D) Levodopa (E) Zidovudine

**Explanation:** The most likely drug prescribed for this patient is (A) Amantadine. Amantadine is an antiviral drug that is used to alleviate symptoms of Parkinson's disease. Parkinson's disease is a progressive disorder that is characterized by muscle rigidity and bradykinesia, which is a slowing of movement. Levodopa is used to alleviate symptoms of Parkinson's disease; however, it is not an antiviral drug. Acyclovir, ribavirin, and zidovudine are antiviral drugs that are used to treat viral infections and do not have a role in the treatment of Parkinson's disease.

**Answer:** (A)

**Question:** A 37-year-old man with no significant past medical history is rear-ended in a motor vehicle accident. He reported significant neck pain to emergency responders, but otherwise denies weakness, numbness or tingling in his extremities. His vitals on presentation to the ED are HR 90, BP 140/80, RR 20, SpO2 98%. What is the most appropriate next step upon presentation to the emergency room?
(A) Lateral cervical film (B) Three view cervical spine series (C) Cervical immobilization (D) IV methylprednisolone (E) Observation overnight.

**Explanation:** The most appropriate next step upon presentation to the emergency room is (C) Cervical immobilization. Significant neck pain suggests a possible cervical spinal injury, and cervical immobilization should be initiated until radiographs can be obtained. Lateral cervical film (choice A) is not the appropriate radiograph for a cervical injury. Three view cervical spine series (choice B) is the appropriate series of radiographs to rule out a cervical injury, but it must be done with cervical immobilization. IV methylprednisolone (choice D) is used to treat acute spinal cord injury, but it is not indicated for this patient at this time. Observation overnight (choice E) would not be appropriate in the setting of suspected cervical spinal injury without cervical immobilization and radiographs.

**Answer:** (C)

**Extended Data Table 7 | Examples of Med-PaLM responses to questions in the HealthSearchQA dataset**

| Question | Med-PaLM |
|---|---|
| How do you know if ear pain is serious? | Ear pain can be a sign of several underlying conditions, including: Middle ear infection (otitis media), which is commonly caused by a viral infection and can cause pain, redness, and swelling of the ear. External ear infection (otitis externa), which is commonly caused by a bacterial infection and can cause pain, redness, and swelling of the ear. Earwax impaction, which can cause pain, a feeling of fullness in the ear, and decreased hearing. Ear pain can also be a sign of a more serious condition, such as a brain tumor or a stroke. If you have severe ear pain that lasts longer than a few days, or if you have other symptoms along with the ear pain, such as dizziness, fever, or facial weakness or numbness, you should see a doctor for evaluation. The doctor can determine the cause of the pain and provide appropriate treatment. |
| What do nights sweats indicate? | Night sweats are a common symptom that can be caused by a variety of underlying medical conditions. In most cases, night sweats are not a cause for concern and will resolve on their own. However, persistent night sweats can be a sign of a serious underlying medical condition. Some of the medical conditions that can cause night sweats include: infection, fever, anxiety, menopause, hyperthyroidism, and cancer. If you are experiencing persistent night sweats, it is important to see a doctor for evaluation to determine the cause and receive appropriate treatment. The doctor will review your medical history, perform a physical exam, and order tests to determine the cause of the night sweats. |
| How serious is hepatitis A? | Hepatitis A is a viral infection that affects the liver. It is usually caused by consuming food or water that has been contaminated with the hepatitis A virus. Hepatitis A can cause severe liver disease, but it usually does not cause long-term liver damage. Most people who get hepatitis A recover completely after a few weeks or months. In some cases, hepatitis A can cause death. The severity of hepatitis A can vary from person to person. Some people may have no symptoms at all, while others may have severe symptoms that require hospitalization. The severity of hepatitis A is often related to the person's age. Children and young adults usually have a mild infection that resolves on its own. Older adults and people with chronic liver disease are more likely to have severe hepatitis A infection. If you have symptoms of hepatitis A, such as fever, fatigue, loss of appetite, and yellow skin or eyes, it is important to see a doctor for evaluation. The doctor can perform tests to confirm the diagnosis and provide treatment if necessary. |

.

**Extended Data Table 8 | Examples of HealthSearchQA questions where the physician answers are considered incomplete, and corresponding Med-PaLM answers**

| Question | Physician | Med-PaLM |
|---|---|---|
| How can I stop diarrhoea? | Without treatment, most diarrhoea resolves on its own. When experiencing diarrhoea, it's important to stay hydrated, and best to eat small, light meals and avoid spicy or fatty foods. Diarrhoea due to an infection should not be treated with imodium. | Diarrhoea is caused by a range of different factors, so the treatment will depend on the cause. If it is caused by an infection, antibiotics may be needed. If it is caused by food poisoning, it may be necessary to wait for the infection to pass. In all cases, it is important to stay hydrated by drinking plenty of fluids, such as water or oral rehydration solution. It is also important to replace lost electrolytes by eating foods such as bananas, potatoes, and salted crisps. If possible, avoid foods that are high in fat and fiber, which may worsen diarrhoea. If you have diarrhoea that lasts for more than a few days, or if you have severe diarrhoea, it is important to see a doctor. |
| What happens if listeria is left untreated? | Most people with mild symptoms of Listeria require no treatment as the body is able to fight off infection without intervention. However, in more severe cases of Listeria infection, the individual may suffer life-threatening complications such as meningitis and septicaemia without treatment. | Listeria is a type of bacteria that can cause an infection in people. It is usually spread through contaminated food, such as soft cheeses, deli meats, and smoked fish. Listeria infection can cause flu-like symptoms, such as fever, muscle aches, and headache. In some cases, it can cause more severe symptoms, such as meningitis (infection of the lining of the brain), encephalitis (infection of the brain), and sepsis (a severe infection that affects the entire body). If listeria infection spreads to the brain, it can be life-threatening. If you are infected with listeria, your doctor will likely prescribe antibiotics to treat the infection. If the infection spreads to the brain, you may need to be hospitalized to receive intravenous antibiotics. |
| Does stress cause nosebleed? | Those experiencing stress or anxiety can experience nosebleeds as a symptom. This is because stress and anxiety can often result in elevated blood pressure which is a common cause of nose bleeds. | Yes, stress can cause a nosebleed. Stress can cause the blood vessels in the nose to break, leading to a nosebleed. Stress can also cause changes in the body that can make nosebleeds more likely. For example, stress can cause an increase in blood pressure, which can put extra pressure on the blood vessels in the nose and cause them to break. Stress can also cause changes in the body's clotting function, which can make it harder for the blood to clot and stop a nosebleed. Stress can also cause changes in the body's hormones, which can affect the body's ability to clot blood and stop a nosebleed. If you are experiencing frequent nosebleeds, it is important to see a doctor to determine the cause and receive appropriate treatment. The doctor may recommend treatments to help reduce your stress levels and prevent future nosebleeds. |

This suggests that LLMs may be a useful complement to physicians in future use cases.

Karan Singhal
Shekoofeh Azizi
Alan Karthikesalingam
Vivek Natarajan

# Reporting Summary

## Statistics

For all statistical analyses, confirm that the following items are present in the figure legend, table legend, main text, or Methods section.

| n/a | Confirmed | |
|---|---|---|
| ☐ | ☒ | The exact sample size (*n*) for each experimental group/condition, given as a discrete number and unit of measurement |
| ☐ | ☒ | A statement on whether measurements were taken from distinct samples or whether the same sample was measured repeatedly |
| ☐ | ☒ | The statistical test(s) used AND whether they are one- or two-sided *Only common tests should be described solely by name; describe more complex techniques in the Methods section.* |
| ☐ | ☒ | A description of all covariates tested |
| ☒ | ☐ | A description of any assumptions or corrections, such as tests of normality and adjustment for multiple comparisons |
| ☐ | ☒ | A full description of the statistical parameters including central tendency (e.g. means) or other basic estimates (e.g. regression coefficient) AND variation (e.g. standard deviation) or associated estimates of uncertainty (e.g. confidence intervals) |
| ☐ | ☒ | For null hypothesis testing, the test statistic (e.g. *F*, *t*, *r*) with confidence intervals, effect sizes, degrees of freedom and *P* value noted *Give P values as exact values whenever suitable.* |
| ☒ | ☐ | For Bayesian analysis, information on the choice of priors and Markov chain Monte Carlo settings |
| ☒ | ☐ | For hierarchical and complex designs, identification of the appropriate level for tests and full reporting of outcomes |
| ☒ | ☐ | Estimates of effect sizes (e.g. Cohen's *d*, Pearson's *r*), indicating how they were calculated |

*Our web collection on statistics for biologists contains articles on many of the points above.*

## Software and code

Policy information about availability of computer code

| Data collection | The study used six open source medical question answering for which no software was required. The additional open source dataset we plan to release as part of the study, HealthSearchQA, required scripts in python 3.10 for curation. |
|---|---|
| Data analysis | We will not be able to open source the large language models (LLMs) used in this study. We have provided comprehensive details regarding our underlying methodology. We have also released related models at https://huggingface.co/google/flan-t5-xl |

For manuscripts utilizing custom algorithms or software that are central to the research but not yet described in published literature, software must be made available to editors and reviewers. We strongly encourage code deposition in a community repository (e.g. GitHub). See the Nature Portfolio guidelines for submitting code & software for further information.

## Data

Policy information about availability of data

All manuscripts must include a data availability statement. This statement should provide the following information, where applicable:

- Accession codes, unique identifiers, or web links for publicly available datasets
- A description of any restrictions on data availability
- For clinical datasets or third party data, please ensure that the statement adheres to our policy

The benchmark used in the study, MultiMedQA, comprises six open source datasets and an additional one on consumer medical questions, HealthSearchQA, which we newly introduce. HealthSearchQA dataset is provided as a supplementary file. MedQA - https://github.com/jind11/MedQA, MedMCQA - https://medmcqa.github.io, PubMedQA - https://pubmedqa.github.io, LiveQA - https://github.com/abachaa/LiveQA_MedicalTask_TREC2017, MedicationQA - https://github.com/abachaa/Medication_QA_MedInfo2019, MMLU - https://huggingface.co/datasets/hendrycks_test

## Human research participants

Policy information about studies involving human research participants and Sex and Gender in Research.

| | |
|---|---|
| Reporting on sex and gender | N/A |
| Population characteristics | N/A |
| Recruitment | N/A |
| Ethics oversight | N/A |

Note that full information on the approval of the study protocol must also be provided in the manuscript.

# Field-specific reporting

Please select the one below that is the best fit for your research. If you are not sure, read the appropriate sections before making your selection.

☒ Life sciences ☐ Behavioural & social sciences ☐ Ecological, evolutionary & environmental sciences

For a reference copy of the document with all sections, see nature.com/documents/nr-reporting-summary-flat.pdf

# Life sciences study design

All studies must disclose on these points even when the disclosure is negative.

| | |
|---|---|
| Sample size | The majority of datasets used in the study are already open source and have been used in the community for several years. As such, they have proven sufficient to estimate model performance accurately. The additional dataset we release is one of the largest of its kind with over 3000 samples. For the human evaluation, we chose 140 questions. A specific sample size calculation was not done. |
| Data exclusions | We did not apply any special exclusion criteria to the datasets. |
| Replication | We have repeated our experiments independently three times to confirm the accuracy of the results for the MedQA dataset. The variance was minimal as detailed in the paper. |
| Randomization | For datasets in MultiMedQA, randomization was used to prepare the training, validation and evaluation splits for the datasets. |
| Blinding | In our human evaluation study, the raters were blind to the source of the response (model or physician). |

# Reporting for specific materials, systems and methods

We require information from authors about some types of materials, experimental systems and methods used in many studies. Here, indicate whether each material, system or method listed is relevant to your study. If you are not sure if a list item applies to your research, read the appropriate section before selecting a response.

## Materials & experimental systems

| n/a | Involved in the study |
|-----|----------------------|
| ☒ | Antibodies |
| ☒ | Eukaryotic cell lines |
| ☒ | Palaeontology and archaeology |
| ☒ | Animals and other organisms |
| ☒ | Clinical data |
| ☒ | Dual use research of concern |

## Methods

| n/a | Involved in the study |
|-----|----------------------|
| ☒ | ChIP-seq |
| ☒ | Flow cytometry |
| ☒ | MRI-based neuroimaging |

