## [Peer Review File · Nature]

Manuscript Title: Large Language Models Encode Clinical Knowledge

Reviewer Comments & Author Rebuttals

Reviewer Reports on the Initial Version:

Referees' comments:

Referee #1 (Remarks to the Author):

A. Summary of the key results

This paper studied the question-answering (QA) problem in the medical domain using large language models (LLMs). It first constructed an evaluation benchmark for a fair comparison, by collecting existing medical QA datasets and also preparing a new dataset, HealthSearchQA. This HealthSearchQA dataset covers popular medical questions that real users will search for in daily life, which is a very useful resource by itself. The experiment results on all these medical QA datasets demonstrate the promising direction of using LLMs to handle medical QA. Strong performance gains are observed when scaling up the LLMs but human evaluations suggest that scaling alone is not enough. There are some potential issues about factuality, consistency, safety, bias, etc. Therefore, this paper can be viewed as a thought-provoking work to call for future work on this topic using the proposed benchmark.

B. Originality and significance: if not novel, please include reference

It is a nice benchmarking paper — a nice combination of existing datasets, a newly added dataset, and extensive experiments covering different aspects of evaluation, including human evaluation to compare the LLMs' performance with the clinician experts' performance. To my knowledge, this is the first, comprehensive benchmark for medical QA. Inviting clinician experts to participate in the benchmark experiments is an impressive idea — It might be hard for academia people to achieve this. So the results and comparisons shall be considered as a milestone.

C. Data & methodology: validity of approach, quality of data, quality of presentation

Since it is a benchmarking paper, as expected, the LLMs (e.g., PaLM) and the prompting methods (e.g., few-shot, chain-of-thoughts, and self-consistency) used in this paper are not brand new. The major contributions of this paper shall be interpreted as (1) contributing one new dataset, (2) putting together the datasets, LLMs, and different prompting methods and obtaining extensive experiment results, and (3) insightful results and discussions about the limitation and future work.

D. Appropriate use of statistics and treatment of uncertainties

The experiments are well-designed. Confidence intervals/standard deviations are reported in

addition to the average results. The comparisons are all fair. I don't see any obvious flaws.

E. Conclusions: robustness, validity, reliability

The authors draw the conclusions in a fairly conservative way. There is no obvious over-claim. Many limitations and potentials have been elaborated, especially on fairness, bias, and ethics.

F. Suggested improvements: experiments, data for possible revision

The paper looks great to me as a benchmark paper for medical QA.

G. References: appropriate credit to previous work?

References look appropriate.

H. Clarity and context: lucidity of abstract/summary, appropriateness of abstract, introduction and conclusions

I enjoyed the paper reading. It is easy to follow.

Referee #2 (Remarks to the Author):

This paper looks at the extent to which a family of large language models (LLM) encode clinical knowledge. Specifically they examine the previously described LLMs PaLM, Flan-PaLM and introduce a new LLM named Med-PaLM, which they have attempted to align to the medical domain. There is an impressive amount of work in this paper and the experiments are well-designed and thoughtful. This paper makes several important contributions, summarized below:

- The creation of a new benchmark dataset (MultiMedQA) consisting of a variety of datasets that differs in terms of format (Multiple choice vs. free-form), domain (open vs. closed), and source of the questions. MultiMedQA is a combination of six previous open-sourced datasets and an additional dataset called HealthSearchQA introduced by this study.
- This is one of the largest studies evaluating the capabilities of the Large Language Model (LLM) on medical question-answering tasks. This work goes beyond testing LLMs on multiple-choice datasets and evaluates the capability of LLMs on open-ended free-form questions.
- The paper further develops an existing framework for the human evaluation of clinical responses generated from LLMs, which is quite comprehensive.
- The study demonstrates that "instruction prompt tuning," a hybrid soft and hard prompt tuning framework, improves the model's performance on the proposed human evaluation framework. The improvements are substantial along many human evaluation axes.
- The paper reaffirms several of the previous works on LLMs by showing that approaches like self-

consistency, instruction-tuning, and scaling help with achieving better performance on question-answering tasks. Using these approaches, the study reports SOTA performance on MedQA, MedMCQA, and PubMedQA datasets.

- The study proposes a measure of uncertainty based on self-consistency that seems promising for assigning confidence to an LLM prediction in multiple-choice questions.

Overall this is an impressive body of work, however, there are some limitations. The most pressing is the lack of evaluation of Med-PaLM on the question answering tasks. Intuition would suggest that the domain specific fine tuning would improve question answering ability, but this was not systematically evaluated. So, we are left in the somewhat incomplete state of knowing that a generic LLM like Flan-PaLM does indeed have a lot of clinical knowledge, but also that the answers it provides for clinical queries are likely insufficient for everyday use due to lack of alignment, which Med-PaLM *does* have. However, as previously stated, we do not know if the alignment process for Med-PaLM in some way diminishes its clinical knowledge as measured by question answering ability.

This key piece of evidence is the major thing lacking from the study in its current form. I provide a more detailed overview of the paper's strengths and weaknesses below:

Strengths:

- The authors demonstrate that Flan-PaLM can achieve SOTA performance on nearly all MultiMedQA tasks, often by a substantial margin. In particular, their best model gets nearly 70% of USMLE practice questions correct, which is a substantial improvement over the previous SOTA.
- The human evaluations are quite convincing that instruction prompt tuning is effective in aligning Flan-PaLM with the proposed axes of evaluation
- The work is thorough for the most part in presenting many evaluations, ablations, and examples.
- The authors state a commitment to open-source the MultiMedQA dataset at the time of publication.

Limitations:

- The analysis with Med-PaLM on multiple-choice questions is incomplete. The study describes the evaluation of instruction fine-tuned Flan-PaLM on only MedQA in the appendix. It seems that the instruction prompt tuning was done separately for the multiple-choice questions for the evaluation on MedQA. So, the soft prompts differ from those used for the human evaluation framework. It would have been interesting to see how the instruction prompt tuning for human evaluation impacts the model performance on MedQA datasets, especially since it was limited to learning the soft prompts, and the goal was to align the model to the medical domain (I would expect same model to continue to do well on multiple-choice questions as well, but the performance of Med-PaLM used for free-form questions on these multiple-choice questions dataset are not reported.)
- There should be some analysis of how many of the questions appear in the text used to pretrain the base PaLM model. Some of these questions have been available for years and it is highly likely

that they appear verbatim in the training data. This is important not only for this study, but for others who wish to use the MultiMedQA to evaluate LLMs.

- The human evaluation framework uses a single clinician to evaluate the model's output along the evaluation axes. Given the small size of the evaluation dataset (140 questions), this is not a very rigorous evaluation of the model. Further, it is not clear if the doctors and non-experts who evaluated the outputs of the model and clinicians were themselves evaluated for their ability to rate the outputs along the specified evaluation axes.

- The study uses various techniques to achieve the best performance on different benchmarks. While self-consistency leads to the best results on MedQA, there is a drop in accuracy on MedMCQA and PubMedQA when using self-consistency. While the authors are transparent about trying various approaches, it raises questions about the robustness of approaches like self-consistency. Since there is no way to know which technique will work in a specific setting or specific datasets, it is somewhat improper to report the best results for each dataset based on trying several approaches to evaluate the model on a specific dataset.

- In section 4.1, it is mentioned that the number of decodes for self-consistency was fixed at 11. It is not clear how this number was arrived upon.

Chain-of-Thought (CoT) prompt examples for MedQA have "We refer to Wikipedia articles on medicine for help." in all the explanations. Does this sentence help in CoT reasoning?

- The technical contributions of this work are somewhat limited. This work uses the PaLM and its instruction-tuned variant Flan-PaLM to test clinical reasoning capability using previously proposed ideas like self-consistency and hard and soft prompting strategies. Further, most of the datasets proposed in this study exist except the HealthSearchQA.

Author Rebuttals to Initial Comments:

Response to Reviewers

Reviewer#1

This paper studied the question-answering (QA) problem in the medical domain using large language models (LLMs). It first constructed an evaluation benchmark for a fair comparison, by collecting existing medical QA datasets and also preparing a new dataset, HealthSearchQA. This HealthSearchQA dataset covers popular medical questions that real users will search for in daily life, which is a very useful resource by itself. The experiment results on all these medical QA datasets demonstrate the promising direction of using LLMs to handle medical QA. Strong performance gains are observed when scaling up the LLMs but human evaluations suggest that scaling alone is not enough. There are some potential issues about factuality, consistency, safety, bias, etc. Therefore, this paper can be viewed as a thought-provoking work to call for future work on this topic using the proposed benchmark.

We thank the reviewer for the constructive and positive feedback. We are grateful that the reviewer recognized the value of the work. The reviewer's positive feedback truly validated the effort.

Reviewer #2

Overall this is an impressive body of work, however, there are some limitations. The most pressing is the lack of evaluation of Med-PaLM on the question answering tasks. Intuition would suggest that the domain specific fine tuning would improve question answering ability, but this was not systematically evaluated. So, we are left in the somewhat incomplete state of knowing that a generic LLM like Flan-PaLM does indeed have a lot of clinical knowledge, but also that the answers it provides for clinical queries are likely insufficient for everyday use due to lack of alignment, which Med-PaLM *does* have. However, as previously stated, we do not know if the alignment process for Med-PaLM in some way diminishes its clinical knowledge as measured by question answering ability. This key piece of evidence is the major thing lacking from the study in its current form.

We appreciate the reviewer's thoughtful and detailed feedback, which helped us to strengthen the quality of the work. We are glad to see that the reviewer appreciated our contributions, including MultiMedQA, detailed human evaluation, and model alignment experiments. We have provided detailed responses to comments below. Most importantly, we have included **new experiments** to address the "key piece of evidence" the reviewer wished to see; the experiments demonstrate that the alignment process does not significantly diminish clinical knowledge for Med-PaLM, as measured by multiple-choice accuracy.

The analysis with Med-PaLM on multiple-choice questions is incomplete. The study describes the evaluation of instruction fine-tuned Flan-PaLM on only MedQA in the appendix. It seems that the instruction prompt tuning was done separately for the multiple-choice questions for the evaluation on MedQA. So, the soft prompts differ from those used for the human evaluation framework. It would have been interesting to see how the instruction prompt tuning for human evaluation impacts the model performance on MedQA datasets, especially since it was limited to learning the soft prompts, and the goal was to align the model to the medical domain (I would expect same model to continue to do well on multiple-choice questions as well, but the performance of Med-PaLM used for free-form questions on these multiple-choice questions dataset are not reported.)

We thank the reviewer for this great suggestion. We have run additional experiments to study how instruction prompt tuning on longform questions in the medical domain may affect performance on multiple-choice questions in MultiMedQA. Specifically, we evaluated the same Med-PaLM model used in the human evaluation results on each of MedQA, MedMCQA, PubMedQA, and MMLU clinical topics datasets. Note that this model was not trained using these datasets. Despite this, as the reviewer suggested, the model continued to do well on these questions, performing about the same as Flan-PaLM. Intuitively, this suggests that instruction prompt tuning is able to align the model to the requirements of consumer medical

question answering (as measured by our human evaluation) without significantly affecting its clinical knowledge. We have detailed this analysis in **Appendix A.5** as a case study.

There should be some analysis of how many of the questions appear in the text used to pretrain the base PaLM model. Some of these questions have been available for years and it is highly likely that they appear verbatim in the training data. This is important not only for this study, but for others who wish to use the MultiMedQA to evaluate LLMs.

We thank the reviewer for the suggestion here. We ran additional analyses to understand the possibility of overlap and model memorization for both the multiple-choice questions and the longform consumer questions in MultiMedQA.

We computed overlap statistics between the PaLM training corpus and MultiMedQA multiple-choice evaluation questions (MedQA, MedMCQA, PubMedQA, MMLU clinical topics) using a sliding-window approach with a length of 25 words (based on the average length of questions in the benchmark). We count any multiple-choice question with at least one overlapping window as overlapping, even if the training corpus omits the answer to the question. We observe limited overlap using this conservative approach, suggesting that this does not significantly affect our results. The per-dataset statistics are detailed in **Appendix A.1**.

For the longform consumer questions in the MultiMedQA, individual questions are generally short commonly-asked health questions, so overlap with the web-scale training corpus used for PaLM models is expected. To understand the possibility of model memorization affecting Med-PaLM's outputs, we took the full set of model responses used in human evaluation for this work (140 answers to questions in HealthSearchQA, MedicationQA, LiveQA) and used a sliding-window approach with 25 words. We checked each window of 25 words against the training corpus, and found no matches for any question. This indicates Med-PaLM is not directly memorizing answers to these common health questions from the PaLM training corpus. This analysis is described in **Appendix A.1**.

The human evaluation framework uses a single clinician to evaluate the model's output along the evaluation axes. Given the small size of the evaluation dataset (140 questions), this is not a very rigorous evaluation of the model. Further, it is not clear if the doctors and non-experts who evaluated the outputs of the model and clinicians were themselves evaluated for their ability to rate the outputs along the specified evaluation axes.

We appreciate the reviewer's feedback. We would like to clarify that we did not rely on a single clinician only, and that the evaluation was conducted with single-labels from a diverse panel of nine clinicians. We also employed 14 separate axes of evaluation per question to permit a more

nuanced evaluation than has hitherto been reported in the literature for AI systems performing longform open-ended medical question-answering. Furthermore, we went to additional lengths, in terms of time and cost, to ensure the inclusion of clinician experts in addition to laypeople, with labelers based in the US, UK and India to provide diversity in perspectives and lived experience. As Reviewer #1 noted, “Inviting clinician experts to participate in the benchmark experiments is an impressive idea — It might be hard for academia people to achieve this.” As per the study, most prior works in the field only report objective metrics and do not include any human evaluation at all. We appreciate the reviewers’ comment regarding whether “*clinicians were themselves evaluated to rate the outputs along the evaluation axes*”. The guidelines were developed and iterated using sequential processes of focus groups as described in the manuscript, followed by rounds of iteration of triplicate assessment of questions in order to converge labeller performance.

We hope to conduct further reader studies to assess the efficacy of our model in the real world in future work. We provide an extensive discussion of potential future directions for work on inter-clinician variation in evaluation of model performance in Section 6 (“Limitations”), most particularly noting that such work will require further translational steps to narrow the use-case and real-world setting in which such evaluations are performed.

The study uses various techniques to achieve the best performance on different benchmarks. While self-consistency leads to the best results on MedQA, there is a drop in accuracy on MedMCQA and PubMedQA when using self-consistency. While the authors are transparent about trying various approaches, it raises questions about the robustness of approaches like self-consistency. Since there is no way to know which technique will work in a specific setting or specific datasets, it is somewhat improper to report the best results for each dataset based on trying several approaches to evaluate the model on a specific dataset.

We thank the reviewer for bringing up this point. We would like to clarify that accuracy only drops for PubMedQA when using self-consistency, not MedMCQA. In general, we observed higher variability in results on the PubMedQA dataset across different approaches, which is likely due to its relatively small sample size (500 questions). We furthermore find the “ground truth” in PubMedQA to be somewhat “noisy” or open to debate between experienced clinician-scientists when examined on a per-case basis; further explaining the potential for minor variations in performance when this dataset is used as a single benchmark. One supporting factor is the human performance on the PubMedQA dataset of 78.0% [Jin et al. 2019]. Since our models are performing roughly at this mark, it is likely that label noise is a significant factor in any differences in results.

We’ve also updated the caption for Figure 3 to clarify that we report the best result across methods in that figure, for full transparency. Elsewhere in Section 4 (“Results”), we report results per method, so readers are able to see the effect of individual techniques.

In section 4.1, it is mentioned that the number of decodes for self-consistency was fixed at 11. It is not clear how this number was arrived upon. Chain-of-Thought (CoT) prompt examples for MedQA have “We refer to Wikipedia articles on medicine for help.” in all the explanations. Does this sentence help in CoT reasoning?

We have experimented with different numbers of decodes for self-consistency and observed a similar performance by using a larger number of decodes. For example, in Figure 5, we ran 41 decodes for selective prediction on the MedQA dataset and obtained an accuracy of 67.4%, comparable to 67.6% obtained with 11 decodes. The choice of 11 decodes for all other datasets is to allow all experiments to finish within a reasonable time without consuming excessive computational resources.

Regarding the prompt, we do not observe a clearly significant difference if that sentence is omitted (we omitted it in other experiments), and in fact in Appendix A.6 we can see that the model can perform well even with prompting outside of the medical domain. This finding suggests that it’s most important to provide examples of the general format / style of these multiple-choice tasks, instead of the exact details of the prompt.

The technical contributions of this work are somewhat limited. This work uses the PaLM and its instruction-tuned variant Flan-PaLM to test clinical reasoning capability using previously proposed ideas like self-consistency and hard and soft prompting strategies. Further, most of the datasets proposed in this study exist except the HealthSearchQA.

We thank the reviewer for their feedback, and would also like to highlight our technical contributions. We build on the PaLM family of models and existing prompting techniques, but **we also combine them in novel ways to introduce instruction prompt tuning, a data and parameter-efficient alignment technique extremely well-suited for the medical domain.** In contrast, other alignment methods such as Reinforcement Learning with Human feedback are not particularly sample efficient and difficult to implement in practice.

This technique resulted in Med-PaLM, which significantly improved on Flan-PaLM as compared to clinicians across 14 axes of human evaluation. In this work we also provide a basis for further technical research, **as with our proposal to use the distribution of self-consistency decodes as a useful measure of uncertainty.**

Additionally, HealthSearchQA provides a unique view of medical information questions from real-world consumers, which may not be reflected in other datasets.

Lastly, we believe our unique combination of comprehensive benchmarking, rigorous human evaluation, and prompting / model adaptation techniques in addition to state of the art

performance on datasets such as MedQA (USMLE) adds up to a work greater than its individual parts. Or in the words of Reviewer 1, **'the results and comparisons shall be considered as a milestone.'**

We would like to thank the reviewers again for their suggestions. They have significantly improved our work and we hope our responses have addressed all questions.

Reviewer Reports on the First Revision:

Referees' comments:

Referee #1 (Remarks to the Author):

I think the authors have actively addressed the concerns raised by reviewer 2. I'm generally happy with the revision and glad to see it got published soon.

Referee #2 (Remarks to the Author):

I thank the authors for their very comprehensive set of responses. I am satisfied and would be hard pressed to think of additional experiments that could be done to further strengthen the results.